# EINOPS: CLEAR AND RELIABLE TENSOR MANIPULATIONS WITH EINSTEIN-LIKE NOTATION

**Alex Rogozhnikov**[*]
alex.rogozhnikov@ya.ru

## ABSTRACT

Tensor computations underlie modern scientific computing and deep learning. A number of tensor frameworks emerged varying in execution model, hardware support, memory management, model definition, etc. However, tensor operations in all frameworks follow the same paradigm. Recent neural network architectures demonstrate demand for higher expressiveness of tensor operations. The current paradigm is not suited to write readable, reliable, or easy-to-modify code for multidimensional tensor manipulations. Moreover, some commonly used operations do not provide sufficient checks and can break a tensor structure. These mistakes are elusive as no tools or tests can detect them. Independently, API discrepancies complicate code transfer between frameworks. We propose `einops` notation: a uniform and generic way to manipulate tensor structure, that significantly improves code readability and flexibility by focusing on the structure of input and output tensors. We implement `einops` notation in a Python package that efficiently supports multiple widely used frameworks and provides framework-independent minimalist API for tensor manipulations.

## 1 INTRODUCTION

Deep learning (DL) over the past decade achieved a significant progress in analysis and synthesis of images, audio and text (Bengio et al., 2017; Aggarwal et al., 2018; Foster, 2019). Tools relying on these methods became a commodity in production data pipelines.

Available research-and-production frameworks for DL, such as pytorch (Paszke et al., 2019), tensorflow (Abadi et al., 2015), mxnet (Chen et al., 2015), jax (Bradbury et al., 2018), and others, vary in numerous aspects, but their core functionality is built around efficient computations on $n$-dimensional arrays (tensors for brevity[1]). API exposed by frameworks for tensor operations follow the same approach, that combines high efficiency (specialized hardware can be utilized) and user convenience: computations can be expressed in high-level languages, such as Python, using a limited number of exposed and usually pre-compiled operations.

Due to growing usage of DL in production and rising complexity of models, it becomes increasingly more important to provide programming interfaces that enable reliable and scalable development. We demonstrate that approach to define tensor operations taken by existing frameworks does not encourage writing code that is easy to interpret, maintain or modify; additionally, some of the core operations do not conduct sufficient checks and can lead to hard-to-catch mistakes. To address these problems, we propose Einstein-like notation for operations, called `einops`. We implement this approach in a Python (Van Rossum & Drake, 2009) package to allow simple integration of notation into existing code across a variety of frameworks.

**Outline** We first briefly describe *mainstream approach for tensor operations* and point to its issues with examples. We review previously proposed ideas to resolve mentioned problems in *related works*. Our approach, *einops* – a verbose notation for tensor manipulation – is introduced next, followed by the code examples in *case studies*. We implement the notation in a Python package, and explain main design choices in *implementation details* section. We conclude with a *discussion* of `einops` role and common criticisms.

---

[*]Currently at Herophilus, Inc.

[1]Our sincere apologies to readers with backgrounds in mathematics and physics for possible confusion.

## 2 MAINSTREAM APPROACH FOR TENSOR OPERATIONS

Nowadays tensor programming is dominating in DL and playing a crucial role in scientific computing. It first appeared in APL (Iverson, 1962), was popularized by MATLAB (Matlab, 1993) and was spread in Python community by numpy (Harris et al., 2020), and now is provided by multiple frameworks. Currently, tensor manipulations in all mainstream frameworks are based on the following assumptions:

- Tensor is an $n$-dimensional array with shape (e.g. $4 \times 6 \times 3$) and data type (e.g. float32).
- When elements are matched across tensors, axes are aligned by order. Conventions regulate cases of different dimensionalities, e.g. broadcasting in Numpy (2021).
- Multiple operations act differently on axes. Those operations either have conventions about axes order (e.g. recurrent units, convolutions; convolutions expect input to be either BHWC or BCHW ordered) or should be provided with indices of axes that should be treated separately:[2]

```
y = x.max(axis=1)      # some operations are provided with indices
y = x.swap_axes(0, 1)  # of specially treated axes
```

The mixed option is possible when operation has defaults for "special axes" that can be overridden during a call.

Benefits of the mainstream approach are simple API and a baseline implementation, as well as versatility: a lot of research code operates solely using this kind of operations for number crunching. The drawbacks are absence of semantics in an operation and no way to reflect expected input and output. To predict output after a chain of manipulations a researcher/engineer has to carefully keep in the memory (or in the code comments) shape and contents of each intermediate tensor and thus keep track of every operation.

In this listing a batch of images is collapsed into a single image by placing images in a row next to each other

```
# im_stack has shape [b, 3, h, w]
y1 = im_stack.transpose(2, 0, 3, 1).reshape(h, b * w, 3)
y2 = im_stack.transpose(2, 3, 0, 1).reshape(h, b * w, 3)
```

One of y1, y2 contains a correct image, while the other is irrelevant – it requires some time to figure out which is what. A typical computation chain contains multiple tensor operations (only two in this example) and requires writing down all intermediate steps to "debug" the code. Annoyingly, resulting tensors y1 and y2 have the same shapes and mistake in the code may stay under the radar for a long time since the code *never* errors out. The lack of stronger checks is a weak point of conventional operations.

In most cases we also cannot meaningfully visualize intermediate layers, so there is no way to narrow down searches for a problem source. Thus, a researcher/engineer has to check all the code after a failure.

Traditional operations restrict code flexibility: any change in the shape agreements between parts of the code is hard to align and all related code fragments (frequently located in different places) should be updated simultaneously. In the next code fragment we add omitted batch dimension to the code from the vision permutator (Hou et al., 2021), and then update code to support depth:

```
# pytorch-like code without batch dimension, as in the paper
x_h = x.reshape(H, W, N, S).permute(2, 1, 0, 3).reshape(N, W, H*S)
x_h = proj_h(x_h).reshape(N, W, H, S).permute(2, 1, 0, 3).reshape(H, W, C)
# with batch dimension
x_h = x.reshape(B, H, W, N, S).permute(0, 3, 2, 1, 4).reshape(B, N, W, H*S)
x_h = proj_h(x_h).reshape(B, N, W, H, S).permute(0, 3, 2, 1, 4).reshape(B, H, W, C)
# with batch and depth dimension
x_h = x.reshape(B, H, W, D, N, S).permute(0, 4, 2, 3, 1, 5).reshape(B, N, W, D, H*S)
x_h = proj_h(x_h).reshape(B, N, W, D, H, S).permute(0, 4, 2, 3, 1, 5).reshape(B, H, W,
    D, C)
```

---

[2]Pseudocode in the paper corresponds to numpy unless otherwise stated. There is no way to write cross-framework code. This problem we partially address with proposed einops notation.

Modifications are very error-prone: all indices should be recomputed and order of `reshape` operands should be verified. Uncommonly, this fragment is quite self-documenting since final `reshape` in each line hints the intended order of axes after the transposition. In DL code it is common to use `x.transpose(0, 3, 1, 2)` expecting other users to recognize a familiar pattern or leaving a comment.[3] Related, transposition in the code requires operating with *three contexts* in mind (original order of axes, permutation, result order), and even when simplified to just permutation, it's unclear if permutation (`2 0 3 1`) is inverse to (`1 3 0 2`).

Less critical, but still notable source of mistakes is 0-based enumeration of axes (Julia, MATLAB and R use 1-based), while we propose the framework not relying on axis numeration at all. Common to refer to axis with index 0 as "first axis/dimension" (see e.g. numpy documentation), and there is no way to change this habit and avoid confusion between human communication and code.

Finally, common tensor operations can easily break the tensor structure. For example, `reshape`, a common operation in the DL code, easily breaks the tensor structure because a whole tensor is treated as a sequence and no connection is assumed between axes in input and output, see Figure 1.

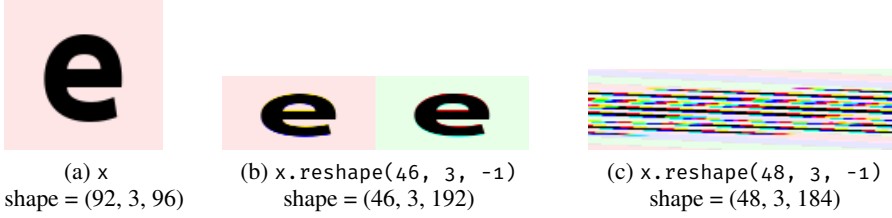

(a) `x`
shape = (92, 3, 96)

(b) `x.reshape(46, 3, -1)`
shape = (46, 3, 192)

(c) `x.reshape(48, 3, -1)`
shape = (48, 3, 184)

Figure 1: Example of breaking the tensor structure in 3d tensor. We use non-convential order HCW for visalizations: (a) original image; (b) partial mixing results in new colors; (c) mixing of all axes. Result shapes are shown underneath.

Aforementioned implementation mistakes can result in rejecting valuable research or drawing incorrect conclusion about the data. Moreover, there is no recipe to identify incorrect implementation, because the result tensor shape and data type are not affected by an error. Even the presence of an error can be obscured: poor performance can be misattributed to a number of other factors: data, hyperparameters, method, etc. All pointed issues have catastrophic importance for research costs and timeline in a setting where one experiment requires dozens to thousands of GPU-hours.

A number of recently proposed architectures (Wu et al., 2021; Tolstikhin et al., 2021; Jumper et al., 2021; Liu et al., 2021; Hou et al., 2021; Touvron et al., 2021) demonstrate that conservative approach with fixed ordering of axes (like BCHW for convolutional networks) is not sufficient. Existing frameworks carry this imprint that does not help with searches for new architectures.

## 3 RELATED WORKS

A commonly taken approach to increase reliability is assigning names/labels to tensor dimensions (below we refer to them as labeled tensors). Most known implementations are `xarray` (Hoyer & Hamman, 2017), `labeled_tensor` in tensorflow (Christiansen & Hoyer, 2016), namedtensor (2019), and named tensors in Pytorch (2019). Despite being around for many years, this idea got little adoption, and not used in the DL code: development stopped for `labeled_tensor` and `namedtensor`, and named tensors in pytorch are still experimental.

In this approach, operations match tensor axes based on labels (common choice of label is string name), and rely on axis label instead of axes order, e.g.:

```
# x1 has axes (x, y, height) and x2 has axes (time, x, y)
x1 + x2 # result has axes (x, y, height, time), maybe in a different order
x1.mean('height') # reduce over axis named 'height'
```

---

[3]Out of 50 last usages of `torch.permute` in Python on GitHub on 22 Nov 2021, only in 4 cases code had comments about result shape, in other cases no information was provided in any other form, including `reshape`.

Axes matching rules vary across implementations. However, we can describe some common issues why this approach did not gain wide support in the DL community:

- Operations focus on modified axes, and neither describe input nor output; a user has to remember axes labels for each of intermediate tensors.

- Less control over data layout: order of axes may significantly influence speed (Weber & Goesele, 2017), but is not transparent.

- Names should be strictly defined and mistakes in names or their alignment between modules may result in the wrong computations, not an exception, e.g. for `namedtensor` package:

```
# x1 has axes (height, width)
# x2 has axes (h, w)
# x3 has axes (heigt)
x1 + x2 + x3 # has axes (height width h w heigt) in some order
```

- Adoption requires a strong buy-in decision, as all code should be axis label-centric. In contrary, transition to labeled tensors in isolated code pieces *only* (e.g. functions) does not prevent (more elusive) errors in interaction between functions, but introduces constant conversions between the code styles. Third-party components need wrapping to support labels.

- Symmetric and antisymmetric tensors with multiple identical axes pose an additional challenge: all axes labels should be unique to allow matching and axis specification.

- Labeled tensors have issues with integration of DL blocks (details in Appendix A).

Proposed implementations of labeled tensors also break the common principle of decomposition in software engineering, which states that every function has its *own* scope with input names and names for intermediate variables. Everything that is shared between scopes should be described in the function signature. Whenever an internal structure of passed or returned entity should be shared between scopes, a type/class/interface/protocol is introduced to describe a passed argument. However, the concept of labeled tensor breaks this logic: it is assumed that called and calling functions agree on axes names, but no way to document these expectations is proposed.

Alternative approach to increase readability and reliability of tensor-operating code is to deliberately set interface restrictions only on large neural modules such as language models, encoders or decoders, as in NeMo Kuchaiev et al. (2019). While allowing to reuse and wrap existing code to glue large components, this approach does not improve internals of the modules where problems are harder to locate and code is less documented. These improvements of high-level interfaces still have their challenges, for example a language model can expect to manipulate sequences of letters and thus expects axis "letter". However, surrounding code may try to use it for prediction of words, pixels or phonemes. Thus, relabeling of axes may be required to "connect" subsystems.

In 2011, M. Wiebe introduced an operation `einsum` in `numpy` package. With some simplifications (absence of covariant and contravariant indices, contracted dimension may be not repeated) `einsum` mimics Einstein summation rule commonly used in physics. `numpy.einsum` stands out from the rest of `numpy` API and for a long time rarely was mentioned in tutorials. However, function universality and expressiveness were beneficial, and it was ported to other packages: tensorflow, pytorch, mxnet, chainer Tokui et al. (2019), etc.

```
numpy.einsum('ij,jk->ik', A, B) # matrix multiplication
numpy.einsum('ijk->ij', C)      # sum over last axis
numpy.einsum('ij,ji->', A, B)   # trace of matrix product
```

`einops` revisits and expands this approach with an emphasis on complex tensor layouts and rearrangements, additional checks and broader functionality. In our work we try to align interface with `einsum` to allow smooth simultaneous usage. However, interface adjustments (such as support for multi-character axes names) are necessary. Most of our changes can be readily applied to `einsum`. Detailed discussion of differences between `einops` and `einsum` is given in Appendix B.

There is an ongoing research to create languages for low-level definition of tensor operations with explicit indexing, e.g. Tensor Comprehensions (Vasilache et al., 2018) and Dex (Paszke et al., 2021).

## 4  einops

Einstein-Inspired Notation for OPerationS, einops, is our proposal to address problems mentioned in Section 2. The core of our approach is a new notation for transformation patterns and, in its basic view, this notation enumerates the tensor elements in one-to-one-correspondence to the set of axes. As seen in examples below, we allow number of axes to be different from tensor dimensionality. The notation uses simple syntax, formally defined in Appendix C, and is based on the following rules:

- axis present only in the input (the left hand side) is reduced (e.g. with max-reduction)
- axis present only in the output is "repeated" (tensor values are the same for all values of new axes)
- all axis identifiers on either side of expression should be unique

Examples of einops transformation patterns are

```
'b c h w -> b h w c' # transpose
'b c h w -> b c'     # reduce on h, w
'b c -> b c h w'     # repeat on h, w
'(h1 h2) (w1 w2) c -> (h1 w1) h2 w2 c' # split image to patches, stack them
```

Each tensor pattern ensures one-to-one mapping between element's positions in the tensor and values of axis variables. This requires all axes used in one tensor pattern to be different (thus traces, permitted by einsum, are not provided by einops). This also requires that ellipsis can be used only once within a tensor pattern.

The main novelty of our notation is the composition and decomposition of axes denoted by parenthesis. (De)composition uses C-ordering convention, intuitively associated with digits in a number:

```
# x is of shape (10, 10, 10, 10), then x[6, 2, 4, 9] == y[6249]
y = rearrange(x, 'a b c d -> (a b c d)')
```

Changing the rightmost of "digits" changes composed index in "small steps", while any change in leftmost results in "large steps", even when axes are not decimal:

```
# Rearrange pattern that composes 3 axes into one: i1 i2 i3 -> (i1 i2 i3)
# Let original array have a shape of [2, 3, 2], result has a length 2x3x2=12
i1                  0  0  0  0  0  0  1  1  1  1  1  1
i2                  0  0  1  1  2  2  0  0  1  1  2  2
i3                  0  1  0  1  0  1  0  1  0  1  0  1
final position      0  1  2  3  4  5  6  7  8  9 10 11
```

Reverse pattern (i1 i2 i3) -> i1 i2 i3 uses the same bijection to decompose an axis into three. Since axis can be decomposed in multiple ways (e.g. 12 could be represented as $2 \times 3 \times 2$ or $1 \times 12 \times 1$ or $3 \times 1 \times 4$, etc.), additional axis size specifications are required during decomposition. The following rule is helpful for C-ordered arrays (default ordering in all current backends): in case the order of axes does not change, result of rearrange is still a view of the original data. For example, rearrangement '(a b c) (d e f) (g h) -> a b (c d) e (f g h)' returns a view and no copy is required.

Axes can be referred to by their size. These anonymous axes have the same role as named, but due to the lack of name they can't be matched across different tensors. Unitary axes have a special meaning – they do not correspond to an axis variable, and thus their behavior is separate from anonymous axes.

```
'h w -> h w 1'       # add unitary axis
'h w -> h w 3'       # repeat values along new anonymous axis
'1 h w 3 -> h w'     # remove unitary axis and reduce on anonymous axis of length 3
'... h w -> ... w h' # transpose two last dimensions
'b ... c -> (...) b c'  # compose all but first and last dimensions
                        # and move resulting new axis to front
'b c ... -> b c'     # reduce on all but first two dimensions
```

All anonymous axes are treated as different even if they share length. Similar to einsum, ellipsis works as a wildcard for zero or more axes, their number and lengths are derived from input shape(s).

Ellipses are not named (grammar allows adding names later), thus all ellipses within a transformation pattern refer to the same group of unnamed axes.

Pattern composition and decomposition became particularly helpful to leverage existing operations for data of higher dimensionality. E.g. if an attention function accepts tensors `k`, `q`, `v` of shape `[batch, seq, channel]`, one can turn it into multi-head attention for 3-dimensional data by composing height, width and depth to a single dimension, and grouping head and batch dimension to ensure independent processing of attention heads: `b h w d head c -> (b head) (h w d) c`. Likewise, other neural blocks can be "upgraded" by rearranging inputs and outputs.

`einops` provides three functions (`rearrange`, `reduce`, and `repeat`) which are shown below in examples (additional axes specifications are provided with `**kwargs`):

```python
# organize 16 images in 4x4 grid
rearrange(im, '(b1 b2) h w c -> (b1 h) (b2 w) c', b1=4, b2=4)
# max-pooling with kernel of size 2x2
reduce(im, 'b c (h h2) (w w2) -> b c h w', 'max', h2=2, w2=2)
# 2x upsampling of individual image by repeating pixels
repeat(im, 'h w c -> (h h2) (w w2) c', h2=2, w2=2, c=3)
```

While all patterns could be handled by a single function instead of three, we made an explicit choice to separate scenarios of "adding dimensions" (repeat), "removing dimensions" (reduce) and "keeping number of elements the same" (rearrange). This helps in producing more specific error messages when a wrong pattern is passed.[4]

In `einsum`, when an axis is present in all tensors, operation performs independently for all values of this axis, which is principle in `einops`. A tensor pattern only identifies correspondence between axes and element's position in tensor, but does not affect arithmetic operation; this ensures that input and output patterns can be changed independently. In particular, `rearrange` is an arithmetic identity (same value returned), but usage of different input and output patterns turns it into a rather universal tool for changing tensor shape/layout.

Proposed notation addresses different problems of the mainstream approach:

- Both input and output are described in the operation definition: tensor dimensionality and expected order of axes. This makes `einops`-based code more declarative and self-documenting. A user is not required to remember or infer shapes of tensors after every operation.

- Input is checked for a number of dimensions and divisibility of corresponding dimensions. The length of dimension is checked if provided.

- A tensor structure cannot be broken by design, because the notation connects input axes (or their constituents) to output axes.

- Axis enumeration is not used, so no way to make one-off mistake.

- Users do not need to compute permutation of axes, those are computed from a pattern.

- `einops` notation alleviates the need to track a tensor layout with patterns.

- `einops` and `einsum` "document" inputs and outputs, simplifying inference of tensor shapes from the code for other tensors that are not direct input/output of `einops`, but are interacting or computed using direct inputs/outputs (an example is given in Appendix E).

We show versatility of `einops` by expressing common numpy (np) operations[5] in Listing 1

```
1  np.transpose(x, [0, 3, 1, 2])          rearrange(x, 'b h w c -> b c h w')
2  np.reshape(x,                          rearrange(x, 'h w c -> (h w) c ')
3    [x.shape[0]*x.shape[1], x.shape[2]])
4  np.squeeze(x, 0)                       rearrange(x, '() h w c -> h w c')
5  np.expand_dims(x, -1)                  rearrange(x, 'h w c -> h w c ()')
6  np.stack([r, g, b], axis=2)            rearrange([r, g, b], 'c h w -> h w c')
7  np.concatenate([r, g, b], axis=0)      rearrange([r, g, b], 'c h w -> (c h) w')
```

---

[4]Post-factum we can confirm this choice: search over Github shows that rearrange, the most restrictive on possible patterns, also accounts for the majority of usages.

[5]Some examples use list inputs, details see in Appendix B.

```
8  np.flatten(x)                              rearrange(x, 'b t c -> (b t c) ')
9  np.swap_axes(x, 0, 1)                      rearrange(x, 'b t c -> t b c')
10 left, right = np.split(image, 2, axis=1)   rearrange(x, 'h (lr w) c -> lr h w c', lr=2)
11 even, odd = x[:, 0::2], x[:, 1::2]         rearrange(x, 'h (w par) -> par h w c', par=2)
12 np.max(x, [1, 2])                          reduce(x, 'b h w c -> b c', 'max')
13 np.mean(x)                                 reduce(x, 'b h w c ->', 'mean')
14 np.mean(x, axis=(1, 2), keepdims=True)     reduce(x, 'b h w c -> b () () c', 'mean')
15 np.reshape(x, [-1, 2]).max(axis=1)         reduce(x, '(h 2) -> h', 'max')
16 np.repeat(x, 2, axis=1)                    repeat(x, 'h w -> h (w 2)')
17 np.tile(x, 2, axis=1)                      repeat(x, 'h w -> h (2 w)')
18 np.tile(x[:, :, np.newaxis], 3, axis=2)    repeat(x, 'h w -> h w 3')
```

Listing 1: Correspondence between `numpy` and `einops` operations.

## 5 CASE STUDIES

We again consider fragments from the vision permutator (Hou et al., 2021). Two examples below only differ in what axis is mixed.

```
# vision permutator - mixing along h
x_h = x.reshape(B, H, W, N, S).permute(0, 3, 2, 1, 4).reshape(B, N, W, H*S)
x_h = proj_h(x_h).reshape(B, N, W, H, S).permute(0, 3, 2, 1, 4).reshape(B, H, W, C)
# vision permutator - mixing along w
x_w = x.reshape(B, H, W, N, S).permute(0, 1, 3, 2, 4).reshape(B, H, N, W*S)
x_w = proj_w(x_w).reshape(B, H, N, W, S).permute(0, 1, 3, 2, 4).reshape(B, H, W, C)
```

While *four* operations were updated simultaneously, in `einops` counterpart changes are limited to swapping `h` and `w` before and after projection, because layouts of input and output are disentangled and changes in one do not propagate to the other. And to remove e.g. batch dimension, one just removes axis `b` from patterns (this is also a generic property of `einops`).

```
# einops: mixing along h
x_h = rearrange(x, 'b h w (n s) -> b n w (h s)', s=S)
x_h = rearrange(proj_h(x_h), 'b n w (h s) -> b h w (n s)', s=S)
# einops: mixing along w. We swapped h and w before and after projection
x_w = rearrange(x, 'b h w (n s) -> b n h (w s)', s=S)
x_w = rearrange(proj_w(x_w), 'b n h (w s) -> b h w (n s)', s=S)
```

The next fragment is derived from OpenAI's implementation of Glow (Kingma & Dhariwal, 2018). As other neural flows, Glow includes multiple rearrangements.

```
def unsqueeze2d(x, factor=2):
    assert factor >= 1
    if factor == 1:
        return x
    shape = x.get_shape()
    height = int(shape[1])
    width = int(shape[2])
    n_channels = int(shape[3])
    assert n_channels >= 4 and n_channels % 4 == 0
    x = tf.reshape(
        x, (-1, height, width, int(n_channels/factor**2), factor, factor))
    x = tf.transpose(x, [0, 1, 4, 2, 5, 3])
    x = tf.reshape(x, (-1, int(height*factor),
                       int(width*factor), int(n_channels/factor**2)))
    return x
# same in einops, no function introduced
rearrange(x, 'b h w (c h2 w2) -> b (h h2) (w w2) c', h2=factor, w2=factor)
```

As for the original implementation, function name is confusing and non-descriptive. To reflect the actual transformation, a name should be changed to `rearrange_by_squeeze_channels_and_unsqueeze_h_and_w`. `einops` alleviates necessity to introduce a function, as arguments describe input, output and the transformation itself. This

example also demonstrates how `einops` performs transformations, as under the hood it makes the same sequence of transformations (reshape-transpose-reshape) as in the original code. Reverse rearrangement (also used in Glow, see Appendix F) requires detailed analysis before implementation as no arguments are shared between original and reverse rearrangements. In `einops` a pattern can be reversed by swapping input and output parts of the pattern.

In Appendix E we analyze and rewrite a larger fragment of code: a multi-head attention module derived from the popular implementation (Huang, 2018).

## 6  IMPLEMENTATION DETAILS

We implement proposed notation in a Python package `einops`. [6]

**Support of multiple frameworks.** `einops` supports a diverse set of DL frameworks (pytorch, tensorflow, chainer, jax, gluon) as well as frameworks for tensor computations: numpy, cupy (Okuta et al., 2017). We refer to them as backends. The major challenge in the support of multiple backends is the absence of common API: even simple operations like repeat, view, or transpose are defined inconsistently.

```
np.transpose(x, [2, 0, 1])          # numpy
x.transpose(2, 0, 1)                # numpy
tf.transpose(x, [2, 0, 1])          # tensorflow
K.permute_dimensions(x, [2, 0, 1])  # keras
mx.nd.transpose(x, (2, 0, 1))       # mxnet
x.permute(2, 0, 1)                  # torch

rearrange(x, 'h w t -> t h w')      # einops (any backend)
```

This inconsistency makes projects like `einops` more valuable for users, as they minimize framework specifics when users do not need it. Proposal (Data-Apis, 2021) may help in convergence on shared API and simplify development of cross-framework applications, including `einops`.

**Backend recognition** in `einops` does not use wrapped imports, which are commonly used in Python to handle optional dependencies:

```
def is_numpy_tensor(x): # einops does not use this approach
    try:
        import numpy as np
        return isinstance(x, np.ndarray)
    except ImportError as _:
        return False
```

Instead, `einops` keeps dictionary that maps a tensor type to a backend. When the type is not found in a dictionary, a pass through backends is done, and before importing the module, `einops` confirms that the backend was *previously* loaded in `sys.modules`, since `einops` will not receive tensors from non-imported modules. The main reasons for this strategy are: a) some backends take a lot of memory and time to load (e.g. tensorflow), which may result in an unforeseen usage of resources if a backend is installed but not used; b) it is possible that a backend is installed incorrectly (e.g. a wrong binary) and import drives to non-catchable segmentation faults; c) checking backends one-by-one incurs unnecessary overhead, which we skip by dictionary lookup.

`einops` has shared logic for parsing and conversion into operations for every framework. This conversion is very efficient: every `einops` pattern is converted into at most 4 tensor operations by backend, even if dimensionality of tensor is large.[7] Overall, `einops` adds marginal overhead on top of the DL frameworks, see Appendix G.

**Support for backends**. Each backend is represented by a proxy class that provides a minimal set of operations necessary to manipulate tensors. In addition, several supplementary methods are required and implemented to allow generic testing: most of tests can be run against any backend. In addition,

---

[6]Einops package is available online at https://github.com/arogozhnikov/einops
[7]There are unavoidable exceptions: some backends cannot reduce multiple axes at once.

`einops` implements **layers** for backends with layers support. This codebase heavily relies on stateless operations and has some backend-specific code.

**Caching** plays an important role in `einops` high performance (Appendix H). There are two layers of caching: i) cache for a pattern and provided axes; ii) cache for a pattern, provided axes, and input shape. When the same pattern is applied to a new shape, only shape verification and computation of unknown axes sizes is done. When a pattern is applied to input of the same shape (quite typical for iterative algorithms, and very common in DL), `einops` only executes sequence of commands with cached parameters.

**Exceptions** are detailed to provide information about performed operation including pattern and provided axes sizes to simplify debugging.

## 7 DISCUSSION

Speaking of criticism, `einops` is sometimes described as "stringly-typed". We should point that it does not make `einops` less reliable compared to the mainstream approach:

```
numpy.transpose: [ndarray, Tuple[int]] -> np.ndarray
rearrange: [ndarray, string] -> ndarray, (ndarray can be polymorphic)
```

So input and output types are not any different except for string not being tuple. In frameworks tensor types do not describe number of dimensions, and existing type system cannot set restrictions on shape components. Static analysis is identically ignorant to mistakes in patterns and e.g. in axes order. There is no static check that prevents users from passing tuple with wrong number of components, or repeats, or just ridiculously large numbers. Interestingly, `numpy.einsum` can accept axis indices not string pattern, but it makes operation less expressive and almost never used.

`einops` does not enforce any alignment of shapes/axes across operations, thus does not provide integrated analysis/tracking of shapes. Unfortunately, there is currently no design that can accomplish this goal either: different flavors of labeled tensors remain prototypes. A successful design should satisfy multiple requirements that are rather challenging and seem to be irresolvable without a new language (or new language features). On the other hand, to become practically valuable, the system should not be too sophisticated. If resulting solution is significantly harder to integrate compared to running every module with several test inputs, it is unlikely to get a wide adoption.

We performed an initial exploration of notation applicability in 2018 when `einops` was open-sourced by reimplementing a set of diverse fragments from popular DL implementations across different tasks and domains (see details in Appendix I). Since its initial release, adoption of `einops` steadily grows, it is currently used in more than 1000 public Github projects including the contributions from widely known AI laboratories. This important evidence (discussed in Appendix J) confirms design choices and ultimate suitability of notation for needs of research in deep learning.

`einops` notation is not tied to a specific implementation or even language: it can be implemented in more efficient languages.[8] Implementation also can use lower-level primitives and optimizations provided e.g. by TVM and MLIR, not framework-provided abstractions. Both directions have a potential to boost `einops` performance. Alternatively, notation can be introduced to lower-level primitives.

## 8 CONCLUSION

We introduce `einops` – a minimalist notation for tensor manipulations that leverages patterns for describing structure of input and output tensors. `einops` mitigates a number of issues common to the conventional tensor manipulation routines, represents a number of commonly used functions with a small API, and makes code more readable and reliable. We implement notation in a Python package that provides identical API across a variety of tensor frameworks. Cross-framework support, expressiveness of code, additional checks and simple integration into existing projects make `einops` a convenient tool for researchers and engineers.

---

[8]`einops` was independently ported to Rust language https://docs.rs/einops/latest/einops/

ACKNOWLEDGMENTS

We thank M. Cramner, C. Garcia, E. Glazistova, A. Grachev, D. Ivanov, J. Henriques, S. Hoyer, T. Likhomanenko, A. Molchanov, S. Ramakrishnan and others for participating in user studies and/or discussions of notation and API. Multiple improvements to the project documentation were made by C. Garcia and other community members.

We thank Phil Wang for publishing numerous high-quality implementations of new architectures using `einops` and introducing practitioners to the notation by demonstrating its usage in complete projects.

We are grateful to A. Chernyavskiy for feedback on initial draft of this manuscript and to T. Likhomanenko for multiple suggestions on the paper, help during review process, and funding initial stages of this project.

We thank anonymous reviewers and program chair for discussion that helped us to improve and strengthen the paper.

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

## A    Discussion of support for common DL blocks in labeled tensors

While most of labeled tensors packages were targeted at deep learning, none of them proposed integration of DL-specific blocks like convolutions. For illustrative purpose let us discuss different possible scenarios how labeled axes could be added to convolution:

- Should convolution stick to existing behavior and rely on axes order not names? If so, we get no additional checks, but create potential problems since axes names are required in other operations (e.g. reductions), but users should drop the names before every convolution, and reorder axes if necessary. Next, users have to set axes for convolution output, in agreement with input axes labels.

- Should convolution rely on axes names? If so, should axes names be fixed? For instance, convolution may require input to have dimensions ("channel", "height", "width") while all the other dimensions could be considered "batch-like" and preserved in the output? In this case, almost all the tensors will have the same axes names, thus name checks become meaningless. If one decides to apply attention or e.g. recurrent unit, each switch would require renaming axes.

- Should output names be predefined? Thus all tensors produced by convolutions would have the same labels, still preventing meaningful checks. That seems to produce more potential confusion than help.

- Should input axes names be propagated to output? If yes – which names should be propagated? Output channels at first seem to be disconnected from input ones. However, in practice opposite is equally frequent: in depth-wise convolution each input has one-to-one correspondence with output. Setting up common rules for propagation of "height" and "width" labels meets similar questions and exceptions: compare `padding='same'` and `padding='valid'`.

Similar questions arise for other "building blocks" of DL.

The broader problem is to come up with a concept of axis "label" that is indeed helpful in development, i.e. sets restrictions that help in detecting or preventing coding errors, while introducing limited performance and coding overhead.

All design choices taken (including support of various DL blocks) should be in agreement with each other, guided by some high-level logic that works for multiple models and applications. This is hardly feasible without a well-defined concept of axis "label", which was not proposed by implementations.

## B    Einops and einsum

`einops` notation is largely inspired by `numpy.einsum` and extends it for single-tensor transformations: in Listing 1 only 2 operations of 17 (transpose and swap axes) can be implemented with `einsum`. We design interfaces in `einops` to be aligned with `numpy.einsum` for simultaneous usage. In the case study of multi-head attention, Appendix E, we demonstrate how they should interact together. Overall, `einops` implementation deviates from `numpy.einsum` in the following:

- `einops` supports arbitrary reductions (max-, min-, sum-, mean-, logaddexp-reductions as well as callables to support any other reduction) while `einsum` can be used to perform sum-reduction only.

- While `einsum` allows only one-character names for axes, `einops` supports multi-character ones: digits, underscore, and arbitrary capitalization are allowed, while space is used as a delimiter. For example, in `einsum` pattern `'hw c -> c hw'` transposes 3-dimensional tensor, while this is an operation on 2-dimensional tensors in `einops`.

- While `einsum` implementations suffer from non-aligned support for spaces between names and capitals, `einops` notation is backend-independent.

In addition to these discrepancies, we introduce several new concepts in `einops` which are absent in `numpy.einsum`:

- *Composition and decomposition of axes*, a new core functionality described in Section 4.

- *Unitary axes*: in any pattern a user can use `1` to reflect axes of size one.
- Specification of axes size and verification of shapes and divisibility.
- *Anonymous axes*: axes that are present only on lhs or rhs can be specified with just size in the pattern. For example: `repeat(x, 'h w -> h w c', c=3)` can be `repeat(x, 'h w -> h w 3')`. Note that unitary axes are not a sub-case of anonymous axes.
- *Introduction of new axes* with `einops.repeat` by repeating elements along the axis. We demonstrate that this operation allows `einops` to support repeats and tiles.
- *List inputs*. `einops` accepts a list of tensors with identical shapes and dtypes, which are stacked. For example, `rearrange([r, g, b, alpha], 'channel h w -> h w channel')`. This extends `einops` to support tensor stacking and concatenations.
- *Layers*. `einops` functions have layer counterparts, e.g. `einops.layers.torch.Rearrange('b c h w -> b (c h w)')` can be used inside `Sequential` as one of the modules.[9]

and at the same time we do not implement some features of `numpy.einsum`:

- *Axes repeats on lhs*. While `einsum` allows writing e.g. `'ii->'` to get a matrix trace we dropped this in `einops` to prevent coding mistakes.
- *Multi-tensor transformations*. While there is an `einops` layer that operates on two tensors, the core functionality of `einops` used by most users is one-tensor transformations. However, `einops` novelties (new axes, anonymous axes, unitary axes, specification of axes sizes) can readily be applied to extend `einsum`.
- *Implicit indexing*. This feature relies on characters sorting, does not encourage descriptive axes names as those define output axes order, and makes output implicit, not explicit. Implicit indexing also cannot be applied to `einops.reduce` and `einops.repeat`. There is a negligible usage ($< 1\%$) of list-inputs syntax in `numpy.einsum`, and it is not added to `einops`.

## C   FORMAL GRAMMAR OF EINOPS PATTERNS

To accommodate differences, covered in appendix B, but still keep similarity with `numpy.einsum`, `einops` patterns follow a formal grammar defined in Listing 2. This grammar significantly evolved from the prototype stage of `einops` notation, when we performed a user study described in Appendix D.

```
transformation_pattern = tensor_pattern, '->', tensor_pattern ;
tensor_pattern = [delimiter], axis_group, {',', delimiter, axis_group}, [delimiter] ;
axis_group = axis | "(", [delimiter], ")"  | ( [delimiter], axis, {delimiter, axis},
    [delimiter] ) ;
axis = '...' | unitary_axis | anonymous_axes | named_axis ;
unitary_axis = '1' ;
delimiter = ' ', {' '} ;
named_axis = <any python identifier that does not start or end with underscore> ;
anonymous_axis = <natural number greater than one> ;
```
Listing 2: Extended Backus–Naur Form grammar of `einops` pattern notation for a single tensor.

## D   CONFIRMATION OF READABILITY AND INTUITIVENESS

At the time of the first `einops` prototype, the idea of using operations with named axes tied to operation itself not to a tensor was not around within DL. Besides, among DL frameworks only pytorch had reasonable support for `einsum` (with multiple performance issues reported).

That is why at the prototype stage we developed a test to probe several design choices and to answer the following questions:

- Is the notation easy to pick up?

---

[9]In examples from pytorch documentation (otherwise `Sequential`) modules are implemented using custom "forward" to accomplish this transformation with `x.view` or `x.flatten`.

- Does the notation require any specific introduction?

- Can users recognize operations without a context?

Specifically, 8 subjects with sufficient (>1 year) exposure to tensor programming in at least one of the DL frameworks were asked to complete the following questionnaire: subjects were not provided with any description of function, or implementation, or other context.[10]

```python
# All imports are ignored. What is the output of each print statement?
x = torch.zeros(10, 20, 30, 40)

y = transpose(x, 'bhwc->bchw')
print(y.shape) # Q1

y = transpose(x, 'bhwc->bc(hw)')
print(y.shape) # Q2

y = transpose(x, 'bhw(c,h1,w1)->b(h,h1)(w,w1)c', h1=2, w1=2)
print(y.shape) # Q3

y = transpose(x, 'b(h,h1)(w,w1)c->bhw(h1w1c)', h1=2, w1=2)
print(y.shape) # Q4

y1, y2 = transpose(x, 'bhw(c,g)->gbhwc', g=2)
print(y1.shape, y2.shape) # Q5

y = transpose(x, 'b1sb2t->b1b2st')
print(y.shape) # Q6

# operator @ is matrix multiplication
t = transpose(x, 'bchw->(bhw)c') @ torch.rand(20,50)
print(t.shape) # Q7

y = transpose(t, '(bhw)c2->bc2hw', b_hw=x.shape)  # first operand is t
print(y.shape) # Q8

y = transpose(t, '(bhw)c2->bc2hw', b=30, h=10)
print(y.shape) # Q9
```

This user study had some additional ideas that were rejected later: comma as a delimiter within parenthesis (non-parenthesized commas conflict with `numpy.einsum`) and parsing of multiple shape arguments (e.g. `b_hw=x.shape`). Even some participants who correctly answered corresponding questions (5 participants of 8 in both cases) reported they had initial confusion with these features.

Only two participants were aware about `einsum` function but not used that in their coding practice. Questions (Q1, Q2, Q6, Q7, Q9) were correctly answered by all participants. This confirms that notation based on named axes is intuitive and can be picked up without a special introduction. Post-questionnaire interview showed that participants could describe a particular context when an operation (e.g. Q1, Q3) could be used.

While single-character variables names (with an optional digit) did not cause confusion, we have replaced this due to conflicts with potential future features (e.g. unitary axes). Function name `transpose` was also changed to `rearrange` to avoid conflicts with existing operations in different DL frameworks.

However the real value of design choices in `einops` can be confirmed in a long time-scale and within large codebases, settings that can't be captured by small user studies.

---

[10]Some participants had no experience with pytorch and they were informed that `x.shape = [10,20,30,40]`.

# E   CASE STUDY: MULTI-HEAD ATTENTION

Below we provide a shortened implementation of multi-head attention based on Huang (2018). For brevity, we remove weights initialization and mask support.

```python
class ScaledDotProductAttention(nn.Module):
    def __init__(self, temperature, attn_dropout=0.1):
        super().__init__()
        self.temperature = temperature
        self.dropout = nn.Dropout(attn_dropout)
        self.softmax = nn.Softmax(dim=2)

    def forward(self, q, k, v):
        attn = torch.bmm(q, k.transpose(1, 2)) / self.temperature
        attn = self.softmax(attn)
        attn = self.dropout(attn)
        output = torch.bmm(attn, v)

        return output, attn

class MultiHeadAttention(nn.Module):
    def __init__(self, n_head, d_model, d_k, d_v, dropout=0.1):
        super().__init__()
        self.n_head = n_head
        self.d_k = d_k
        self.d_v = d_v

        self.w_qs = nn.Linear(d_model, n_head * d_k)
        self.w_ks = nn.Linear(d_model, n_head * d_k)
        self.w_vs = nn.Linear(d_model, n_head * d_v)
        self.attention = ScaledDotProductAttention(temperature=np.power(d_k, 0.5))
        self.layer_norm = nn.LayerNorm(d_model)
        self.fc = nn.Linear(n_head * d_v, d_model)
        self.dropout = nn.Dropout(dropout)

    def forward(self, q, k, v):
        d_k, d_v, n_head = self.d_k, self.d_v, self.n_head

        sz_b, len_q, _ = q.size()
        sz_b, len_k, _ = k.size()
        sz_b, len_v, _ = v.size()

        residual = q

        q = self.w_qs(q).view(sz_b, len_q, n_head, d_k)
        k = self.w_ks(k).view(sz_b, len_k, n_head, d_k)
        v = self.w_vs(v).view(sz_b, len_v, n_head, d_v)

        q = q.permute(2, 0, 1, 3).contiguous().view(-1, len_q, d_k) # (n*b) x lq x dk
        k = k.permute(2, 0, 1, 3).contiguous().view(-1, len_k, d_k) # (n*b) x lk x dk
        v = v.permute(2, 0, 1, 3).contiguous().view(-1, len_v, d_v) # (n*b) x lv x dv

        output, attn = self.attention(q, k, v)

        output = output.view(n_head, sz_b, len_q, d_v)
        output = output.permute(1, 2, 0, 3).contiguous().view(sz_b, len_q, -1) # b x lq
    x (n*dv)

        output = self.dropout(self.fc(output))
        output = self.layer_norm(output + residual)

        return output, attn
```

Original implementation demonstrates a number of issues that we previously discussed: unchecked and hard-to-track transformations, necessity to keep comments about the shape. Frequent usage of -1 in reshapes makes it simple to introduce errors. The critical part of computations (attention) is offloaded to a separate module, which does not check input and which is not documented: dimensions of inputs and outputs are not defined, just as the connection between them. Thus, provided ScaledDotProductAttention cannot be considered as an independent, self-containing or reusable module.

Inlining of ScaledDotProductAttention inside MultiHeadAttention could improve the situation, but would also make the problem with shapes more obvious, as more comments would be required for inlined variables.

We can compare that with einops implementation, where all computations are done in a single module. Each axis is easy to track throughout the code. An axis index is needed three times (explicitly in softmax, implicitly in fc and layer_norm), but those are easy to find from the code without any additional comments – which demonstrates how einops "implicitly annotates" code in the proximity.

```python
class MultiHeadAttentionNew(nn.Module):
    def __init__(self, n_head, d_model, d_k, d_v, dropout=0.1):
        super().__init__()
        self.n_head = n_head

        self.w_qs = nn.Linear(d_model, n_head * d_k)
        self.w_ks = nn.Linear(d_model, n_head * d_k)
        self.w_vs = nn.Linear(d_model, n_head * d_v)
        self.fc = nn.Linear(n_head * d_v, d_model)

        self.dropout = nn.Dropout(p=dropout)
        self.attn_dropout = nn.Dropout(p=0.1)
        self.layer_norm = nn.LayerNorm(d_model)

    def forward(self, q, k, v):
        residual = q
        q = rearrange(self.w_qs(q), 'b l (h k) -> h b l k', h=self.n_head)
        k = rearrange(self.w_ks(k), 'b t (h k) -> h b t k', h=self.n_head)
        v = rearrange(self.w_vs(v), 'b t (h v) -> h b t v', h=self.n_head)
        attn = torch.einsum('hblk,hbtk->hblt', [q, k]) / np.sqrt(q.shape[-1])
        attn = self.attn_dropout(attn.softmax(dim=-1))
        output = torch.einsum('hblt,hbtv->hblv', [attn, v])
        output = rearrange(output, 'h b l v -> b l (h v)')
        output = self.dropout(self.fc(output))
        output = self.layer_norm(output + residual)
        return output, attn
```

## F SQUEEZE EXAMPLE FROM GLOW WITH REVERSE TRANSFORMATION

Squeeze and unsqueeze implementations are derived from OpenAI's implementation of Glow (Kingma & Dhariwal, 2018).

```
def squeeze2d(x, factor=2):
    assert factor >= 1
    if factor == 1:
        return x
    shape = x.get_shape()
    height = int(shape[1])
    width = int(shape[2])
    n_channels = int(shape[3])
    assert height % factor == 0 and width % factor == 0
    x = tf.reshape(x, [-1, height//factor, factor,
                    width//factor, factor, n_channels])
    x = tf.transpose(x, [0, 1, 3, 5, 2, 4])
    x = tf.reshape(x, [-1, height//factor, width //
                    factor, n_channels*factor*factor])
    return x

def unsqueeze2d(x, factor=2):
    assert factor >= 1
    if factor == 1:
        return x
    shape = x.get_shape()
    height = int(shape[1])
    width = int(shape[2])
    n_channels = int(shape[3])
    assert n_channels >= 4 and n_channels % 4 == 0
    x = tf.reshape(
        x, (-1, height, width, int(n_channels/factor**2), factor, factor))
    x = tf.transpose(x, [0, 1, 4, 2, 5, 3])
    x = tf.reshape(x, (-1, int(height*factor),
                    int(width*factor), int(n_channels/factor**2)))
    return x
```

einops counterparts for both functions can be written as

```
rearrange(x, 'b (c h2 w2) h w -> b c (h h2) (w w2)', h2=factor, w2=factor)
rearrange(x, 'b c (h h2) (w w2) -> b (c h2 w2) h w', h2=factor, w2=factor)
```

where one can readily see from the patterns that the second operation is inverse to the first one.

## G EINOPS PERFORMANCE

To demonstrate that the overhead brought by einops on top of DL framework is negligible we measure performance of several case studies. We compare original pytorch implementations and their einops versions in the following scenarios, see Table 1: CPU or CUDA backends, with enabled or disabled JIT (just-in-time compilation), different input tensor sizes. In our performance benchmark we use einops 0.3.2, pytorch 1.7.1+cu110, CUDA 11.0. We use AWS EC2 p3.2xlarge instance for benchmarks. The average time of the forward pass is measured in milliseconds by IPython's module timeit. For multi-head attention case study, see Appendix E, einops implementation uses pytorch einsum operation which gives additional overhead. That is why, for this case study we consider einops implementation with and without einsum. For the case of unsqueeze, we use popular open-source port to pytorch https://github.com/chaiyujin/glow-pytorch.

From these benchmarks even einops-rich code works on a similar range of speeds as pytorch-only. In particular, unsqueeze2d consists only of einops operation, shows similar speed both on CPU for various input sizes and GPU with larger input sizes. Since computationally most expensive operations are convolutions and tensor products (linear layers, attention), not rearrangements, this overhead will typically have marginal contribution to the result. Thus, attention and permutator are closer to

practical use cases, and demonstrate that difference in performance becomes notable only for small inputs when GPU is used.

Table 1: Performance comparison between original pytorch implementation and its `einops` version for different case studies. Performance is measured in ms via IPython's `timeit` module.

| Case Study | Input Size | Impl. | CPU Backend | | CUDA Backend | |
|---|---|---|---|---|---|---|
| | | | w/o JIT | JIT | w/o JIT | JIT |
| attention | (32, 64, 512) | einops | 21.85±0.04 | 21.13±0.03 | 1.30±0.00 | 1.05±0.00 |
| | | w/o einsum | 21.31±0.04 | 20.99±0.06 | 1.14±0.00 | 0.90±0.00 |
| | | original | 21.09±0.03 | 20.85±0.05 | 1.02±0.00 | 0.71±0.00 |
| | (32, 128, 512) | einops | 48.06±0.06 | 47.37±0.05 | 1.39±0.00 | 1.30±0.00 |
| | | w/o einsum | 46.94±0.04 | 46.28±0.05 | 1.28±0.00 | 1.29±0.00 |
| | | original | 46.87±0.08 | 46.33±0.04 | 1.29±0.00 | 1.28±0.00 |
| | (32, 256, 512) | einops | 150.40±0.18 | 149.53±0.17 | 2.78±0.00 | 2.79±0.00 |
| | | w/o einsum | 146.01±0.20 | 144.96±0.19 | 2.75±0.00 | 2.76±0.00 |
| | | original | 145.19±2.07 | 145.94±0.21 | 2.75±0.00 | 2.75±0.00 |
| | (32, 512, 512) | einops | 495.31±1.19 | 493.90±0.60 | 6.51±0.00 | 6.53±0.01 |
| | | w/o einsum | 488.37±0.93 | 487.16±0.96 | 6.47±0.00 | 6.48±0.01 |
| | | original | 491.20±2.79 | 488.64±0.23 | 6.45±0.01 | 6.46±0.00 |
| permutator | (32, 32, 32, 32) | einops | 7.05±0.01 | 7.03±0.01 | 0.52±0.00 | 0.52±0.00 |
| | | original | 7.08±0.02 | 7.05±0.02 | 0.44±0.00 | 0.40±0.00 |
| | (64, 64, 64, 64) | einops | 237.53±1.67 | 237.05±0.24 | 2.51±0.00 | 2.56±0.00 |
| | | original | 234.57±0.17 | 234.66±0.21 | 2.50±0.01 | 2.54±0.00 |
| unsqueeze2d | (32, 32, 32, 32) | einops | 1.68±0.00 | 1.73±0.00 | 0.07±0.00 | 0.13±0.00 |
| | | original | 1.66±0.00 | 1.72±0.00 | 0.05±0.00 | 0.09±0.00 |
| | (32, 64, 64, 64) | einops | 16.91±0.02 | 17.01±0.04 | 0.14±0.00 | 0.19±0.00 |
| | | original | 16.89±0.03 | 17.14±0.02 | 0.12±0.00 | 0.14±0.00 |
| | (32, 128, 128, 128) | einops | 129.53±0.05 | 129.95±0.10 | 0.71±0.00 | 0.76±0.00 |
| | | original | 129.16±0.16 | 131.44±0.12 | 0.69±0.00 | 0.72±0.00 |

## H ROLE OF CACHING

To estimate the role of caching we conduct a study with very small `numpy` arrays, where we generate $10^4$ patterns performing the same transformation using IPython's `timeit` functionality. Cache can't accommodate all these patterns and runs syntax parsing, validation and inference of dimensions.

```
from einops import rearrange
import numpy as np

patterns = [
    f'i{i} (j k) l m ... -> i{i} j (k l) (m ...)'
    for i in range(10_000)
]

x = np.zeros([2, 2, 2, 2, 2]) # small tensor

# caching mechanism is not used
%%timeit
for pattern in patterns:
    rearrange(x, pattern, j=2)

# caching is in use
%%timeit
pattern = patterns[0]
for _pattern in patterns:
    rearrange(x, pattern, j=2)
```

Outputs for the two cases show more than 10-fold difference:

- without caching: 663 ms ± 20.9 ms per loop (mean ± std. dev. of 7 runs)
- with caching: 37.1 ms ± 165 µs per loop (mean ± std. dev. of 7 runs)

Times were measured on Macbook Pro 2018 using Python 3.9.0, `numpy` 1.20.3 and `einops` 0.3.2.

## I EINOPS FLEXIBILITY AND APPLICABILITY

To confirm wide applicability of `einops`, we search for Github code in pytorch that actively relies on `view`/`reshape`/`transpose`/`permute`. We restrictively consider only popular (>1000 stars) repositories. Official documentation, examples and `torchvision` are also included (all three repositories pass the previous criterion). We sample 16 fragments that cover a diverse set of applications and architectures and also smaller fragments from the same repositories, see Table 2.

Table 2: Comparison between original implementation and rewritten version with `einops` measured in number of code lines and characters.

| Fragment | # Lines | | # Characters | |
|---|---|---|---|---|
| | Original | einops | Original | einops |
| LeNet-like network | 19 | 15 | 626 | 352 |
| Super-resolution | 17 | 11 | 678 | 478 |
| Style transfer | 6 | 3 | 192 | 107 |
| LSTM and language modeling | 15 | 16 | 745 | 753 |
| LSTM token-by-token | 31 | 22 | 1266 | 990 |
| ShuffleNet | 126 | 43 | 4007 | 1932 |
| ResNet | 58 | 42 | 2106 | 1658 |
| FastText | 24 | 8 | 723 | 281 |
| CNN for text classification | 43 | 16 | 1767 | 799 |
| Tacotron | 36 | 23 | 818 | 662 |
| Transformer's attention | 78 | 32 | 2637 | 1538 |
| Self-attention GANs | 34 | 19 | 1506 | 959 |
| Time-sequence prediction | 27 | 25 | 1125 | 1015 |
| Spacial transformer network | 36 | 27 | 1099 | 1012 |
| Highway convolutions | 6 | 6 | 256 | 239 |
| GLOW | 30 | 5 | 917 | 250 |

These examples are rewritten in `einops` and `torch.einsum`: interface of models is kept identical to the original implementation to demonstrate that `einops` does not demand global code adjustments. In addition, introduction of `einops` layers allows to avoid new classes and to reuse `nn.Sequential` in four cases (LeNet, Super-resolution, FastText, ResNet).

Resulting code in `einops` is shorter, see Table 2, while we use fewer method chains and do not pack multiple operations in a single line. Out of 16 fragments, in one case length (measured in number of code lines or in number of characters – results are identical) is higher than in original (LSTM and language modeling), in one case is identical (Highway convolutions), and in all other cases is tangibly lower.[11]

---

[11] Side-to-side comparison between original code and `einops` version with comments can be found online
https://einops.rocks/pytorch-examples.html

## J    EINOPS ADOPTION

When it comes to estimating suitability of notation (or any other tool) to help in research, one can try to dissect this question into multiple small questions: how many different cases are covered by notation? can it be implemented? can it be integrated with existing tools? is resulting code shorter? how fast is resulting code? is modification with notation easier or faster? While we cover in our studies multiple such factors, this "dissection" does not provide holistic view on role and integration into existing workflow and tooling.

`einops` has taken a "test by time" approach. Developed, implemented and open-sourced in 2018, it got adoption in public projects in majority of largest AI laboratories, which is a strong indicator of novelty, usefulness and reliability. For context, pytorch claims to be user-focused Paszke et al. (2019). In keynote talk[12] they confirm poor relevance of micro-benchmarks and rely on adoption as a (lagging) confirmation of design choices.

As of November, 2021, Github reports more than 1000 usages of `einops` by other repositories with increasing frequency (last 100 were published in 18 days). In the majority of cases they implement approaches or DL models that were proposed *after* release of `einops`. Implementations use different DL backends and span different modalities (even non-traditional applications like fluid dynamics).

This inductive validation (if it was applicable to a large number of new cases, it should work in more) is extremely time-inefficient, but provides the most reliable proof. `einops` is used in open source projects from AI laboratories, who use and maintain competing deep learning stacks. This forms another important (though less statistically reliable) indication for adoption.

---

[12]Outline is available at https://soumith.ch/posts/2021/02/growing-opensource/.

