# OpenReview forum: "Einops: Clear and Reliable Tensor Manipulations with Einstein-like Notation"
_ICLR.cc/2022/Conference — ICLR 2022 Oral_

### Official Review · Reviewer_p5Km · 2021-10-17

**Correctness:** 2
**Technical Novelty And Significance:** 2
**Empirical Novelty And Significance:** 2
**Recommendation:** 3
**Confidence:** 4

**Main Review:**

First, this paper studies efficient programming paradigms for multi-dimensional array operation, which is an important intermediate layer for modern ML systems, especially for deep learning systems. Personally I tend to buy these advantages of einops claimed by the author, including:

+ providing semantic check for tensor operations;
+ including high expressiveness and flexibility for tensor manipulating interfaces;
+ supporting multiple backend runtimes efficiently;
+ making tensor manipulating code more readable and reliable, and etc..

These claims sound very promising and valuable as a ML toolkit.

However, there are some fundamental concerns I have for the paper:

+ The writing is problematic as an academic paper. Comprehensively, this paper reads like a technical blog, which tries to introduce and advertise a Python library; some statements in this paper are casual, i.e., in the end of Section 7, the author says "We intentionally omit discussion of user conveniences provided by einops package: anonymous axes, ellipsis, list inputs and neural layers.", although I would not doubt about it if reasonable amount of evidence (e.g., empirical study or analysis) was presented, this sentence sounds a little ungrounded. Further, the organization of the paper can be polished as well. To be specific, I do not get why Section 2 & Section 3 are split, it seems that both sections are talking about the issues and limitations of STOA systems.
+ There is a lack of reasonable amount of empirical study to justify the statements about the contribution of the work. For example, the author claims that einops "significantly improves code readability and flexibility" in the abstract, I would expect some user-study (which could invite a group of participants to accomplish some programming tasks about tenor manipulation with and without einops and measure some objective or subjective metrics for evaluation) to justify this claim---note that this is pretty standard in first-tier PL/SE volumes, it would be easy to find hundreds of such papers and adopt such methodology for evaluation. The current Section 5 can be considered as good motivating examples, but it is not sufficient to support this claim following the principle of scientific study. The efficiency evaluation of the implementation is also missing, e.g., it is important to learn how much overhead is introduced by imposing einops over the backend systems, e.g., what is the runtime gap between einops implementations and direct usage of the backend systems' interfaces.
+ Lastly, I feel lost about the discussion about the difference between einops and numpy einsum. As far as I learn, the interface looks similar (at least from coding examples). As so, I would expect some clear and concrete statements about the distinguish components of einops (different from einsum).

**Summary Of The Paper:**

This paper introduces einops, a tensor manipulating library in Python, to efficiently support multidimensional tensor manipulations widely adopted in deep learning.
Highlighted advantages of einops include:
+ providing semantic check for tensor operations;
+ including high expressiveness and flexibility for tensor manipulating interfaces;
+ supporting multiple backend runtimes efficiently;
+ making tensor manipulating code more readable and reliable, and etc..

**Summary Of The Review:**

This paper introduces an interesting tensor manipulating library in Python, supporting Einstein-notation style operations over multi-dimensional arrays for deep learning. The library manages to run the convenient interface over different popular deep learning frameworks.

However, there is a lack of solid empirical study to validate the effectiveness and efficiency of the design and clear discussion about the difference from the existing tool, i.e., einsum.

---

> ### Author Response · Authors · 2021-11-15
> **Response to reviewer p5Km**
>
> Thank you for the review, p5Km
> > … which is an important intermediate layer for modern ML systems ...
>
> > These claims sound very promising and valuable as a ML toolkit.
>
> We appreciate this evaluation.
>
> > this paper reads like a technical blog, which tries to introduce and advertise a Python library
>
> This is a generic statement that we expect to be detailed, at least accompanied by a set of examples.
>
> Paper states the problem, discusses context and previous approaches, introduces notation, demonstrates proposed solution (package), describes technical challenges and (sometimes unique) decisions that made this solution possible and practical.
> We support our work with analysis of code examples. If there is something that is redundant and does not line up, please be specific and point this out.
>
>
> > "We intentionally omit discussion of user conveniences provided by einops package: anonymous axes, ellipsis, list inputs and neural layers." […] this sentence sounds a little ungrounded
>
> 9 pages is the hard limit, we will rephrase and provide additional information in Appendix.
>
> > To be specific, I do not get why Section 2 & Section 3 are split, it seems that both sections are talking about the issues and limitations of STOA systems.
>
> Section 2 focuses on the currently widely accepted solution that we build upon and integrate into; also we discuss its flaws. However, einops does not completely replace existing operations (see Appendix A).
> In contrary, section 3 describes alternatives to einops, and how those aim to resolve named flaws.
>
> > There is a lack of reasonable amount of empirical study to justify the statements about the contribution of the work. For example, the author claims that einops "significantly improves code readability and flexibility" in the abstract, I would expect some user-study (which could invite a group of participants to accomplish some programming tasks about tenor manipulation with and without einops and measure some objective or subjective metrics for evaluation) to justify this claim---note that this is pretty standard in first-tier PL/SE volumes, it would be easy to find hundreds of such papers and adopt such methodology for evaluation. The current Section 5 can be considered as good motivating examples, but it is not sufficient to support this claim following the principle of scientific study.
>
> Please check the general comment to all reviewers.
>
> >  The efficiency evaluation of the implementation is also missing, e.g., it is important to learn how much overhead is introduced by imposing einops over the backend systems, e.g., what is the runtime gap between einops implementations and direct usage of the backend systems' interfaces.
>
> Thanks for the feedback. For practical use-cases the overhead is negligible. We will add details on this comparison in the revised version.
>
> > Lastly, I feel lost about the discussion about the difference between einops and numpy einsum. As far as I learn, the interface looks similar (at least from coding examples). As so, I would expect some clear and concrete statements about the distinguish components of einops (different from einsum).
>
> We appreciate the feedback and will make this clear in the revised version. Detailed explanation we give in the general comment to all reviewers.

---

### Official Review · Reviewer_ntk4 · 2021-10-21

**Correctness:** 4
**Technical Novelty And Significance:** 3
**Empirical Novelty And Significance:** 2
**Recommendation:** 6
**Confidence:** 4

**Main Review:**

Strengths:
-	The proposed toolbox allows for simple and easy to read code tensor manipulation operations and operates with most popular Python frameworks.
-	The usage of indices names is more flexible compared to already existing ones in the einsum operation in numpy and Pytorch, which are known to have some limitations, e.g. number of dimensions is limited by the number of letters.
-	It is very useful for using Tensor Networks where number of interacting tensors are usually big and can have a large number of dimensions.

Weaknesses:
-	The paper does not provide any advance in theory or new algorithm for machine learning. It is limited to introduce a useful and appealing new coding tool.
-	The paper does not mention its application for computing and manipulating Tensor Networks, missing a very important usage for which there is a growing audience eager to have such convenient tool.
-	EINOPS does not consider operations involving two or more tensors
-	A comparison in terms of computation cost is missing in the paper


**Summary Of The Paper:**

In this paper, a new Python toolbox called EINOPS (Einstein Operations) is introduced. The proposed toolbox allows for applying tensor operations on a single tensor data, such as reshaping, reducing (max, mean, etc), repeating, permuting, and others; in such a way that it improves code readability and flexibility. The paper illustrates about current issues on available solutions, typically in numpy and shows how EINOPS solves those issues. Also, the toolbox supports widely used frameworks such as Pytorch, TensorFlow and others providing framework-independent minimalist API for tensor structure manipulations.

**Summary Of The Review:**

I found the tool very useful for any application involving tensors with Python. The paper is focused on deep learning applications. However, I am sure it is also very useful for dealing with Tensor Networks (TNs), where many core tensors are interconnected. The paper lack of a description of its application to TN contractions, for example.

It seems that current implementation only consider operations on single tensors. I would suggest to include also operations on two or more tensors, similarly to einsum in numpy and Pytorch, but using a more flexible indices labelling.

A comparison with previously available solutions, via numpy or Pytorch, for example, in terms of computation cost is missing in the paper, making it difficult to evaluate if there is some price to pay.

---

> ### Author Response · Authors · 2021-11-15
> **Response to reviewer ntk4**
>
> Thank you for the review, ntk4.
>
>
> > I found the tool very useful for any application involving tensors with Python. The paper is focused on deep learning applications. However, I am sure it is also very useful for dealing with Tensor Networks (TNs), where many core tensors are interconnected. The paper lack of a description of its application to TN contractions, for example.
>
> We agree with your judgement and we agree TNs can benefit from einops, e.g. einops test suite has tensor train as one of the cases.
> Einops + TNs require detailed discussion with examples, but we are unlikely to accommodate this in the paper (and we believe that the right exposure of TN community to einops and analysis of benefits/cases/practices/potential costs is of its own value, and better be done by the experts in TN)
>
> > It seems that current implementation only consider operations on single tensors. I would suggest to include also operations on two or more tensors, similarly to einsum in numpy and Pytorch, but using a more flexible indices labelling.
>
> Thank you for this suggestion.
>
> While not mentioned in the paper, einops has one layer implementing generalized einsum for two tensors, one of which is learnable "weight". It covers a number of practical cases (e.g. MLP, MLP-mixer).
> We demonstrate in examples (page 6 last row and page 7 first row) that stack and concatenate of multiple tensors can be implemented in einops.
> We consider adding generalization of einsum as a part of einops, however inconsistent support by backends delays this feature.
> Other multi-tensor operations are a subject of future research.
>
> > A comparison with previously available solutions, via numpy or Pytorch, for example, in terms of computation cost is missing in the paper, making it difficult to evaluate if there is some price to pay.
>
> Thanks for the feedback. For practical use-cases the overhead is negligible. We will add details on this comparison in the revised version.

---

### Official Review · Reviewer_gRMH · 2021-10-25

**Correctness:** 2
**Technical Novelty And Significance:** 2
**Empirical Novelty And Significance:** 2
**Recommendation:** 3
**Confidence:** 5

**Main Review:**

There are numerous issues with the paper, which I will try to summarize below.

I should mention that, since no code was included in the submission, my assessment of this work is based entirely on the paper and the ideas expressed therein. I am not judging the work based on technical aspects related to the library except for those described in the paper.
However, a complete assessment of the paper should also include a code review (see my first comment in the "Weaknesses" section).

**Strenghts**

1. I am a great fan of einsum notation for expressing tensor operations and I agree with the authors on the value of extending the notation as they did. I could see myself, as a practitioner, using the "rearrange" function (I thought that the example of reshaping an array of images into a 4x4 grid on page 6 was nice).
As an engineering feature, the einops package could be useful.

2. Using the library seems to make the code less verbose in the examples provided in the paper. This gives a reasonable advantage in terms of the readability and maintainability of the code (although this is a purely subjective assessment, as I argue below).

3. The fact that the library supports multiple backends is a plus, although I have some concerns about some design choices discussed in the paper (see below).

**Weaknesses**

Unfortunately, I think that there are considerable flaws in the paper, the motivation behind it, and the described implementation of the library.
I will try to summarize everything in the following points.

1. While I agree that much of machine learning is based on tensor manipulation, this paper is not suitable for publication at ICLR.
Even if we were to focus only on tensor manipulation as a machine learning exclusive, better venues exist for this kind of paper, like the JMLR Open Source Software track. There, the paper could be reviewed along with the code (which was not included in this submission even if it is a crucial part of the authors' contributions).
I realize that libraries like TensorFlow and PyTorch had a paper published at some top conferences like ICLR, but it's clear that their novelty and impact on the community was incomparably larger at the time of publication.

2. The motivation for the paper is weak. I will give some examples of claims made by the authors to motivate einops that I don't think hold up:
    - Users have to remember axes order: this is largely solved by einsum, not einops exclusively.
    - Control over data layout: einops does not solve the issue, it just gives a simpler interface to control it.
    - Einops is declarative and self-documenting: einsum too, einops simply extends the paradigm.

3. The contributions of the paper are not enough to consider a publication.

    The paper should describe the notation in detail, but it doesn't. Instead, it is halfway between a description of the notation and a documentation for the package. The result is that it works badly as both. Some examples:

    - The notation is never formally described, for instance as a grammar, and one must look at examples to infer the form of a correct string.
    - Many features like anonymous axes and ellipsis indexing are essentially ignored, even if they are a key part of the notation.
    - A comparison with einsum is missing, which confuses the reader and does not highlight some features that are present in einsum but not einops (for example, computing the trace and the implicit notation).

    Also, the authors claim that they "align the interface with einsum to allow smooth simultaneous usage" but this is misleading. Einops is clearly inspired from einsum and simply adds some features (while also removing others, as I said above).
In this regard, throughout the paper the authors seem to imply that einops solves problems (like that of rearranging) that were previously unsolvable (e.g., second paragraph of page 6). This is also misleading since einops is merely an interface/API to existing functions.

4. The example on page 4 concerning convolution is unclear.
For example: when convolutions rely on axes order, the user is not "expected" at any time to use named axes for other operations.
It is also not clear what kind of "name checks" would be prevented by assigning fixed names to the axes in convolution, and how einops overcomes this issue.
I might have misunderstood this whole paragraph, so please correct me if I'm wrong.

    In this regard, if the interaction with "neural layers" is a primary motivation for developing the library, the feature should be discussed more in depth (it is only briefly mentioned once, towards the end, with no explanation).

5. There are many non-rigorous/subjective claims. I have found at least five:

    - Footnote 4: "New users frequently continue trying to imagine the tensors layout in memory." -- On what grounds are the authors making this claim without any references? Have the authors conducted user studies?
    - Page 9: "Caching plays an important role in einops high performance" -- Since there are no performance benchmarks, how is performance quantified here? I assume that the performance of einops is equal to or worse than the backend, given the extra overhead. Is it faster than equivalent einsum operations? Can the authors quantify the "important role" of caching?
    - Page 9: "einops was also referred once as a good intermediate solution" -- By whom? How is this whole reported conversation meaningful to the reader?
    - Page 9: "Design of such system is much harder than it sounds: previous ideas failed" -- What does it mean that designing is hard and how can the authors be sure that it wasn't due to their own limitations? Whose previous ideas have failed and in what context?
I am obviously not implying anything about the programming and engineering abilities of the authors, I am merely pointing out that claims like this should not be made in a scientific paper.
    - Page 9: "We observed that einops notation gets picked up for describing tensors with packed dimensions" -- Again, did the authors perform a user study to make this claim? And, in any case, did einops notation get picked up more easily than einsum notation? If so, why?

**Other minor issues**

6. Figure 1: the example assumes that the reshape is done by hardcoding the dimensions and that the programmer will make a typo that "luckily" does not crash. First, we don't know how often this kind of bug happens and, second, this issue can be easily bypassed by accessing the actual shapes (like the authors do on page 6, line 2-3 of the code block at the bottom). I wouldn't use this as a primary motivation for the rearranging function.

7. Page 4, third bullet point: the authors imply that the code should crash, but in fact it is a perfectly well-formed expression that should not crash. Again, the authors should not assume the likelihood of a bug or what the users want to do.

8. Page 7: "Einops alleviates necessity to introduce a function, as arguments describe input, output, and the transformation itself". This is a claim in support of the readability of einops, although it holds for einsum already. While I partially agree, einsum/ops notation could also be seen as less beginner-friendly because it requires the users to know the syntax. For example `arr.flatten()` seems (subjectively) more descriptive than `rearrange("ab->()", arr)`, especially if one doesn't already know einsum notation.

**Comments on the implementation details**

9. The choice of automatically detecting the backend from the input is arguable.
Quoting from the famous Zen of Python by Tim Peters: "Explicit is better than implicit" and "In the face of ambiguity, refuse the temptation to guess."

    For example, a situation could happen in which a Numpy array is given as input to a TensorFlow-based user-defined function.
Since TensorFlow knows to automatically convert arrays to tensors at runtime, this is standard and expected behaviour.
Replacing the initial call of this imaginary function with an einops-based one would result in the first operation happening in Numpy and not TensorFlow.

    I advise the authors to adopt a paradigm like that of multi-backend projects like Keras in which the users decide explicitly what backend to use (through a config file or by making the backend explicit at import time -- e.g., `from einops.torch import ...`).

10. Can the authors provide an example of how they improved the exception message for broadcasting w.r.t. the one from Numpy? How is it easier to understand?

**Suggestions**

After revising the paper, the authors should consider submitting it to an appropriate venue for open source contributions.

The revision should make it more clear that einops is an extension of einsum and should highlight the concrete additions and limitations of the proposed einops.

The authors should make the paper more rigorous: the notation should be formalized, the claims about usability should be removed unless backed up by user studies, and the motivation of the paper/library should be rethought.
The paper could also focus less on examples (which are more suitable for the documentation).

**Summary Of The Paper:**

This paper extends the Einstein summation notation (einsum) of Numpy to introduce some additional features:

- naming axes arbitrarily (i.e., with more than a single character);
- controlling how groups of axes are flattened (called "rearranging" in the paper);
- repeating axes;
- reducing axes with different operations other than the sum.

The notation is implemented in a Python library called "einops" that works with different backends for tensor computation (PyTorch, TensorFlow, Jax, etc.).

**Summary Of The Review:**

**Positive aspects**

- The proposed extension of einsum is useful and I can see it having large adoption in the community of scientific computing.

**Negative aspects**

- I don't believe that ICLR is the correct venue for a paper like this.

- The novelty and quality of the paper are rather limited.

- The motivation for the paper is weak and much of the advantages brought by the proposed "einops" were already brought by the original "einsum" notation.

- The presentation is, at times, confusing.

- There are many non-rigorous claims that cannot be tolerated in a scientific publication.

- There are some design choices that could lead to unexpected problems when using the library.

---

> ### Author Response · Authors · 2021-11-15
> **Response to reviewer gRMH, part 1.**
>
> First of all we want to say thanks to reviewer gRMH for an extremely detailed review. However, we disagree on most points and will delve in details later.
> Specific suggestions and detailed points reviewer gRMH left make it easy for us to improve the paper and/or keep discussion focused.
>
> Global comment:
>
> The review largely misses placement of the paper and its value for ICLR participants, instead considering it as an implementation/documentation thing.
> ML/DL researchers experiments in most cases involve tensor manipulation, and claim that improvement to this process "is not suitable for publication at ICLR" is ... weird.
>
> Reviewer gRMH also pushes on statistical validation, please see our discussion of its insufficiency in comment to all reviewers.
>
> Comments in this review which oppose einsum and einops are mostly irrelevant (see detailed comparison in the comment to all reviewers).
> Historically, einops was open-sourced before proper implementation of einsum appeared in most of the DL frameworks (Example: in TF supported only two arguments, did not support ellipsis, did not support repeated indices; keras, gluon, mxnet did not provide this function at all).
> Einops documentation and examples (one example is given in Appendix A) argue for usage of einsum along with einops, and likely contributed to the process of following einsum adoption, as there is a large intersection between the communities, advocating for usage of these two in DL.
> Einops always cited its inspiration einsum (including paper)
>
> > I realize that libraries like TensorFlow and PyTorch had a paper published at some top conferences like ICLR, but it's clear that their novelty and impact on the community was incomparably larger at the time of publication.
>
> Tensorflow paper claims no novelty and makes insufficient attribution to previous works they build on.
> Pytorch paper focuses on design choices and implementation, and does not claim any novelty.
> Neither of the papers mentions any user studies on API design.
> Tensorflow paper has a list of recommendations based on experience of porting a single deep learning network (inception).
> Pytorch uses a number of arxiv papers (not compared to anything) as a proof of design and adoption.
>
> With all respect, in our paper we claim a novel notation for tensor manipulation (which is different from einsum, see their comparison "einsum and einops") compared to the examples you provided.
>
> Further, pytorch claims to be user-focused. And in a much-later talk (https://soumith.ch/posts/2021/02/growing-opensource/), pytorch lead confirms our points, why and how they "measured" design choices. Pytorch project is at the stage when choices wouldn't be frowned upon, which made such (keynote) talk possible.
>
> > repeating axes;
>
> The way it is written in the review is incorrect, there is no such operation as "repeat axis", but "repeat tensor elements along new axis"
>
> > Users have to remember axes order: this is largely solved by einsum, not einops exclusively.
> > Einops is declarative and self-documenting: einsum too, einops simply extends the paradigm.
>
> Please see a general comment and response to all reviewers. Most of einops operations can't be expressed with einsum, so einsum provides very limited remedy, basically only for sum-reduction and transposition, and only with one-letter names.
>
> > Control over data layout: einops does not solve the issue, it just gives a simpler interface to control it.
>
> So ... this is not valuable? Not useful? Paper states the opposite? Point is missed.
>
> > The notation is never formally described, for instance as a grammar,
>
> Thanks for the feedback. We will add formal syntax in Appendix in the revised version.
>
> >  and one must look at examples to infer the form of a correct string.
>
> Users never reported issues with it.
>
> > Many features like anonymous axes and ellipsis indexing are essentially ignored, even if they are a key part of the notation.
>
> Thanks, we will add details in Appendix in the revised version.
>
> > A comparison with einsum is missing, which confuses the reader and does not highlight some features that are present in einsum but not einops (for example, computing the trace and the implicit notation).
> > Also, the authors claim that they "align the interface with einsum to allow smooth simultaneous usage" but this is misleading. Einops is clearly inspired from einsum and simply adds some features (while also removing others, as I said above).
> > And, in any case, did einops notation get picked up more easily than einsum notation?
>
> We will add a detailed comparison in Appendix. Again, see a general comment and comment to all reviewers.

---

> ### Author Response · Authors · 2021-11-15
> **Response to reviewer gRMH, part 2.**
>
>
> > In this regard, throughout the paper the authors seem to imply that einops solves problems (like that of rearranging) that were previously unsolvable (e.g., second paragraph of page 6). This is also misleading since einops is merely an interface/API to existing functions.
>
> Tensorflow is merely an API to CuDNN/CuBLAS/CuFFT/... functions. Numpy is merely an API to C/FORTRAN functions. Pytorch is (was) an interface/API for lua-torch cpp/cuda backend. OpenBLAS is merely … etcetc. This comment seems to argue with the concept of software stacks.
>
> Second paragraph of page 6 does not claim, or implies, or reads that it was impossible. Exact phrase from the paragraph:  "... particularly helpful to leverage existing operations ...".
>
> Moreover, einops is not just interface, but a novel notation (that was already ported e.g. to rust independently of us).
>
> > The example on page 4 concerning convolution is unclear. For example: when convolutions rely on axes order, the user is not "expected" at any time to use named axes for other operations. It is also not clear what kind of "name checks" would be prevented by assigning fixed names to the axes in convolution, and how einops overcomes this issue. I might have misunderstood this whole paragraph, so please correct me if I'm wrong.
>
> The paragraph discusses issues with integrating named tensors, and this 'the user is not "expected" at any time to use named axes for other operations' just straight claim to not use them anywhere.
>
> Einops does not introduce tensor-attached labels and does not face this problem.
>
> > In this regard, if the interaction with "neural layers" is a primary motivation for developing the library, the feature should be discussed more in depth (it is only briefly mentioned once, towards the end, with no explanation).
>
> Issues of labelled tensors should be resolved by labelled tensors, not einops, which takes completely orthogonal design.
>
> > There are many non-rigorous/subjective claims. I have found at least five:
>
> Thank you for being specific here, we value this.
> We should point out that 4 of them are in the discussion section, and the last one is a footnote.
> We will rewrite statements to highlight observational character.
>
> > Since there are no performance benchmarks, how is performance quantified here? I assume that the performance of einops is equal to or worse than the backend, given the extra overhead. Is it faster than equivalent einsum operations? Can the authors quantify the "important role" of caching?
>
> Thanks for the feedback. For practical use-cases the overhead is negligible. We will add details on this comparison in the revised version.
>
> > Page 9: "Design of such system is much harder than it sounds: previous ideas failed" -- What does it mean that designing is hard and how can the authors be sure that it wasn't due to their own limitations? Whose previous ideas have failed and in what context? I am obviously not implying anything about the programming and engineering abilities of the authors, I am merely pointing out that claims like this should not be made in a scientific paper.
>
> Section 3 discusses an alternative approach (labelled tensors) that was in different ways independently explored by several groups.
> Years after the inception and despite significant informational support they did not reach adoption, and two were already shut.
>
> E.g. Yann Lecun promoted one of these solutions https://twitter.com/ylecun/status/1080974471689687040?lang=en, others are/were included in the most widespread frameworks. Multiple researchers, not only LeCun (see thread) were convinced that more reliable tensor ops are almost ready and need some polishing.
>
> We see this as an important point that einops was adopted, while seemingly reasonable approaches did not pass "testing by time".
> "Much harder than it sounds" is a very mild and tempered way of reflecting the drastic difference between promises/expectations and results of these alternative designs.
> This controversy is a valuable lesson of software/notation design, that highlights the importance and non-obviousness of our work.
> It is unprofessional to refer to our capabilities, specially in the light of discussing work done not by us. During einops design we were aware of design traps of labelled tensors, however the only confirmation one can give to support a claim is reference to systematic work of other researchers that did not succeed - that's what we observe and report.

---

> ### Author Response · Authors · 2021-11-15
> **Response to reviewer gRMH, part 3.**
>
> > Figure 1: the example assumes that the reshape is done by hardcoding the dimensions and that the programmer will make a typo that "luckily" does not crash. First, we don't know how often this kind of bug happens and, second, this issue can be easily bypassed by accessing the actual shapes (like the authors do on page 6, line 2-3 of the code block at the bottom). I wouldn't use this as a primary motivation for the rearranging function.
>
> This issue can't be bypassed, because
> - Resulting construction becomes not readable even for a 4-dimensional tensor. This becomes insanely long when the name of the tensor is not x, but something like attn_logits.
> reshaped = attn_logits.reshape(attn_logits.shape[0], attn_logits.shape[1] * attn_logits.shape[2], head, attn_logits.shape[3] // head), and we did not transpose result yet
> - Reshape of this kind requires having a name for the tensor and can't be applied to intermediate results of other operations.
> - There is nothing "lucky" in not crashing, any permutation of shape elements provides a valid reshape and does not fail. Sufficient to use WH not HW for a non-square matrix. Wide usage of -1 in reshapes and powers of two as size dimensions increases chances of being "lucky".
>
> > Page 4, third bullet point: the authors imply that the code should crash, but in fact it is a perfectly well-formed expression that should not crash. Again, the authors should not assume the likelihood of a bug or what the users want to do.
>
> This recommendation contradicts the established practice of software engineering. Design of programming languages (that reviewer refers to) is largely based on pulling out/discouraging/substituting language features with highest likelihood of bug.
>
> We want to highlight for the reviewer that a single typo changes semantics of all downstream operations without providing any notice. Non-alignment between any two packages/functions in names is not caught, but propagates.
>
> > Page 7: "Einops alleviates necessity to introduce a function, as arguments describe input, output, and the transformation itself". This is a claim in support of the readability of einops, although it holds for einsum already.
>
> See general comment (once again). This operation can't be rewritten with einsum.
>
> > While I partially agree, einsum/ops notation could also be seen as less beginner-friendly because it requires the users to know the syntax. For example arr.flatten() seems (subjectively) more descriptive than rearrange("ab->()", arr), especially if one doesn't already know einsum notation.
>
> Arr.flatten is objectively less descriptive because in numpy and e.g. keras/tensorflow v1 this operation works differently and either returns 1d tensor or 2d tensor depending on the framework.
> In pytorch flatten uses axis indexing that conflicts with standard python range indexing.
>
> > than rearrange("ab->()", arr)
>
> This is not a valid einops operation. In rearrange neither of dimensions can be reduced, while reviewer reduced all (first point in section "4. einops"). Correct analogue of flatten can be found on page 7, second line.
> Once again, einops is not einsum, and does not use one-letter names.
> Pattern's syntax is correct, though.
>
> > The choice of automatically detecting the backend from the input is arguable. Quoting from the famous Zen of Python by Tim Peters: "Explicit is better than implicit" and "In the face of ambiguity, refuse the temptation to guess."
> For example, a situation could happen in which a Numpy array is given as input to a TensorFlow-based user-defined function. Since TensorFlow knows to automatically convert arrays to tensors at runtime, this is standard and expected behaviour. Replacing the initial call of this imaginary function with an einops-based one would result in the first operation happening in Numpy and not TensorFlow.
> I advise the authors to adopt a paradigm like that of multi-backend projects like Keras in which the users decide explicitly what backend to use (through a config file or by making the backend explicit at import time -- e.g., from einops.torch import ...).
>
>
> We do not agree with these suggestions.
>
> Keras's config file to define which framework it will work with is a catastrophe in the long term: two or more systems/models implemented over different backends can't coexist in the same process. Keras didn't live long enough to face these consequences.
>
> User that can work with either numpy, or tensorflow, or pytorch, but not two - design is a clear no go.
> Einops layers use syntax `from einops.layers.torch import ....` because layers (weight management/hooks/etc) are fundamentally incompatible.
>
> The value of writing tensor backend-polymorphic code was long underestimated (e.g. by tensorflow but not jax; and see data-apis consortium that we mention). Introduction of specific functions complicates backend-polymorphic coding.

---

> ### Author Response · Authors · 2021-11-15
> **Response to reviewer gRMH, part 4.**
>
>
> > Since TensorFlow knows to automatically convert arrays to tensors at runtime
>
> Not automatically, the right word is implicitly. "Explicit is better than implicit" if you find Zen to be an argument.
> So, why is w = x @ y @ z, which possesses in this context the same behavior as rearrange, not prohibited in TF?
> TF's design to make explicit operations, but not conversions, is very controversial and neither supported by its predecessors (theano, mxnet), nor by google's jax, nor by any other popular package. Though it makes sense for the production setting when only TF is approved.
>
> Given type hinting, correctness of result and TF-specificity, we don't see this as a concern.
>
> > Can the authors provide an example of how they improved the exception message for broadcasting w.r.t. the one from Numpy? How  is it easier to understand?
>
> Einops complains on:
> - wrong input dimensionality (e.g. reshape doesn't check it)
> - malformed pattern with missing on one side/repeating axes. E.g. incorrect rearrange from this review would receive (after fixing order of arguments):
>
> ```
> Error while processing rearrange-reduction pattern "ab->()".
>  Input tensor shape: (3, 5). Additional info: {}.
>  Identifiers only on one side of expression (should be on both): {'ab'}
> ```
> This error 1) points to exact pattern to search for 2) provides full information about inputs 3) points to exact mismatch between operation semantics and usage 4) even shows some information about parsing: "ab" is a single axis.
>
> - missing information about axes introduced on RHS
> - mismatch on anonymous axes size, mismatch on unitary (length 1) axes
> - mismatch on composite axes size if all components are defined, e.g. (a b c), a=2, b=3, c=4; if only part of them are provided with explicit size (e.g. b and c) - divisibility is checked
> - insufficient information to infer all axes sizes
> - sizes for identifiers not used in the expression
> - multiple syntax/semantic errors (brackets, incorrect identifiers, parenthesized ellipsis on lhs, improper usage of anonymous non-unitary axes, invalid names, double usage of identifier, double usage of ellipsis, etc)
>
>
> ### To sum up:
>
> Review misses value, placement, related works, and historical context of our contribution. We can be blamed for not discussing the historical context in the paper.
>
> There are several misattributions/misinterpretations which result in wrong claims.
>
> Software (implementation/API) remarks and recommendations are questionable and not grounded. Pointed concerns and recommendations were never brought to our attention by community or other experts since einops was open-sourced more than 3 years ago. Einops currently is used by >1000 public repos/projects, and was independently endorsed by multiple core contributors to large OS projects, so API design and implementation were extensively tested by the community.
>
> We accept comments on limited comparison with einsum, insufficiently formal exposure, lack of benchmarks and formal notation.
>
> Comparisons with other frameworks made by the reviewer unexpectedly confirmed strong sides of our work.
>
> Once again, thanks to the reviewer gRMH for being specific and detailed, we value this. We believe review would also benefit from being more weighted and justified.

---

> > ### Comment · Reviewer_gRMH · 2021-11-20
> > **Reply**
> >
> > I thank the authors for their response. It was indeed my goal to help them improve the paper.
> >
> > >considering it as an implementation/documentation thing
> >
> > I consider it neither a documentation paper nor one that introduces the notation, as I have stated before. I agree with reviewer p5Km that the paper "reads like a technical blog". Please refer to my previous comments for a justification of why I think this is not good.
> >
> > Further proof of this is that the authors have relegated the discussion of the notation to the appendix, indicating that they don't see the need for formal discussion in the paper.
> > The same could be said for the comparison with einsum.
> >
> > Also, code review is required for a paper like this and therefore JMLR OSS is a better venue.
> >
> > > Users never reported issues with it.
> >
> > Is the notation a contribution of the paper? If so, asking that the readers infer the notation from the examples is not sufficient.
> > I don't consider this a satisfactory reply to my comment.
> >
> > > Tensorflow is merely an API to CuDNN/CuBLAS/CuFFT/... functions...
> >
> > The impact of these libraries is not comparable to that of einops.
> > Also: how less verbose is PyTorch compared to the CUDA equivalent? How less verbose is einops compared to the PyTorch equivalent?
> >
> > > Example on page 4
> >
> > I still don't get what that passage is supposed to convey. One unclear sentence is:
> >
> > "Should convolution stick to existing behavior and rely on axes order not names? If so, we get no additional checks, but create potential problems since **user is expected to use axes names in other operations** (e.g. reductions), but has to drop the names before every convolution, and reorder axes if necessary"
> >
> > I assume that this discussion about convolution should somehow tie to the "neural layers" mentioned on page 9. If not, how is this discussion related to einops?
> > As I said, this is just unclear to me, I'm not objecting to anything in particular.
> >
> > > Resulting construction becomes not readable
> >
> > The readability is not what is being discussed here.
> >
> > > any permutation of shape elements provides a valid reshape and does not fail
> >
> > Isn't that true also in einops? That example only makes sense if one hardcodes the output shape.
> >
> > > Page 4, third bullet point
> >
> > I had misinterpreted the example, I have re-read it after your reply and I retract that comment.
> >
> > > Given type hinting, correctness of result and TF-specificity, we don't see this as a concern.
> >
> > Discussing the design choices of TF is outside of the scope of my review.
> > If the authors decide to dismiss this issue as a non-concern, I still advise them to highlight this behaviour clearly, because it is in contradiction to the way TF works.
> >
> > > Einops complains on: ...
> >
> > My question was: "Can the authors provide an example of how they improved the exception message for broadcasting w.r.t. the one from Numpy? How is it easier to understand?"
> >
> > ----
> >
> > I think that we could go on unproductively discussing the finer details ad infinitum, which is why I have not replied point-by-point to many additional things that the authors have written in their reply. Overall, I think that none of the following major points was addressed:
> >
> > - JMLR OSS would be a better venue (due to code review).
> >
> > - The paper is neither a good description of the software nor a good introduction to the notation.
> >
> > - The presentation is, at times, confusing.
> >
> > - There are still many non-rigorous claims that cannot be tolerated in a scientific publication (waiting for the authors to update the paper).
> >
> > - Some design choices could lead to unexpected problems when using the library, and the authors chose to dismiss them as a non-concern (which I don't believe counts as a real solution).

---

> > > ### Author Response · Authors · 2021-11-23
> > > **Response to reviewer gRMH**
> > >
> > > Dear reviewer gRMH,
> > > Thank you for your feedback and time. We would like to clarify some of your questions and points as follows:
> > > > JMLR OSS would be a better venue (due to code review).
> > >
> > > Notation is the core of our approach, and the main novelty. This is not the subject of JMLR OSS, but ICLR is the right venue.
> > > We added the proper formal syntax and constraints for the notation in the revised version in Section 4 to make it clear that notation is the main novelty.
> > > On the other hand, notation without implementation, and integration into existing software design stack is not a viable thing.
> > > We made this (quite a significant bunch of) engineering work to make the notation practically useful and universal.
> > > We do not see this a software (or mainly-software) contribution, as well as we don't see this as two independent contributions.
> > >
> > >
> > > > Some design choices could lead to unexpected problems when using the library, and the authors chose to dismiss them as a non-concern (which I don't believe counts as a real solution).
> > >
> > > In the previous discussion:
> > > - We pointed to the correctness of the result.
> > > - We pointed to a remedy (type hinting). We can point to other (e.g. layers).
> > > - We pointed to an existing operation in TF (a@b@c, where "a" and "b" are numpy tensors, and then numpy multiplication will be used, not tf on first multiplication)  that has way higher computational overhead but not alarming anyone.
> > > - We reconfirm our assessment that this hypothetical situation is not an issue.
> > > Positive changes of backend-polymorphic code should not be ignored or omitted in this discussion.
> > >
> > > > The readability is not what is being discussed here.
> > >
> > > We don't see discussion of code that ignores readability as a worthy discussion.
> > > We provided other points in that response too.
> > >
> > > > I assume that this discussion about convolution should somehow tie to the "neural layers" mentioned on page 9.
> > >
> > > It is not, as we pointed in our previous answer.
> > >
> > > > … If not, how is this discussion related to einops?
> > >
> > > We expect that giving discussion around "why this approach did not work for DL code" is valuable for context since no post-mortems or good discussion of issues were published.
> > > Two years ago the most common questions for einops were "how does it compete with labeled tensors (LT)", "can you integrate LT" or "can you integrate into LT".
> > > Our answer "this thing is not complete and it is unlikely that it will ever be" rarely satisfied and long unwrapping was required.
> > >
> > > Strangely, this same review demands a detailed comparison with einsum in the main text, which is not a competitor of einops (we got this question too, but rarely - if a person "speaks" einsum, similarities and differences are rather obvious. If not - this discussion would bring no value at this point. For a novelty statement this may be different, and we'll try to cover that).
> > >
> > > We rearranged the text and gave high-level comparisons with einsum in Sections 2 and 4 while having detailed discussion in Appendix B. Additionally we moved discussion about convolutions and labeled tensors into Appendix A.
> > >
> > > > The impact of these libraries is not comparable to that of einops.
> > >
> > > If impact is high, then is "just wrapping" fine? On well-known examples we pointed to the lack of argument and that "wrapping" is a common practice in software.
> > >
> > > > any permutation of shape elements provides a valid reshape and does not fail
> > > > Isn't that true also in einops? That example only makes sense if one hardcodes the output shape
> > >
> > > No, it is not: change of output order does not have the same consequences in einops. As we pointed, the following reshape can't be written in rearrange.
> > > Assume that we have torch.zeros(H, W, color, time).reshape(W, H, color * time) and analogue with einops rearrange(x, "h w c t -> w h (c t)"). In both cases we "made a mistake" in ordering "w" and "h" but einops does transpose on these axes while in case of pytorch reshape mixing between these axes happens and there will be a mess with the tensor similar to  Figure 1 (c) .
> > > In einops resulting w and h axes still correspond to w and h, as output shape and output pattern suggest.
> > > In addition, users can provide axes sizes to ensure exact dimensionalities of inputs rearrange(x, "h w c t -> w h (c t)", w=W, h=H).
> > >
> > >
> > > > My question was: "Can the authors provide an example of how they improved the exception message for broadcasting w.r.t. the one from Numpy? How is it easier to understand?"
> > >
> > > Einops does not provide broadcasting, so this seems like misinterpretation.
> > > When a user does not use einops and makes a mistake in reshape/transpose/..., in majority of cases after wrong reshape they receive messages about broadcasting - not because broadcasting is used, but because numpy (and other packages) try to apply broadcasting when shapes don't coincide and report broadcasting error (which is an indicator of mistake, but does not point where mistake is).
> > >
> > > We removed in the revised version mention of broadcasting to avoid confusion.

---

### Official Review · Reviewer_c6LK · 2021-11-02

**Correctness:** 4
**Technical Novelty And Significance:** 3
**Empirical Novelty And Significance:** 3
**Recommendation:** 8
**Confidence:** 4

**Details Of Ethics Concerns:**

The reviewer is not aware of any ethical concerns in this paper

**Main Review:**

**Strength of the work.** First of all, the paper provides a handful of case studies with insightful analysis on
user experience on the existing design of operators, and then based on those studies it derives a series
of notations for highly user friendly operator design. By providing various comparison between
existing ones on real-world, the paper leads to a convincing proof of the strength of
the proposed notation for deep learning researchers' daily usecases, in terms of expressiveness,
elegance, usability and debuggability.

Second, the paper comes with solid engineering efforts that offload the computation to underlying
libraries, including NumPy, TensorFlow, PyTorch, MXNet, Keras etc. Effectively, with the solid engineering
effort, the library serves as a meta framework that can be reused across backends with close-to-zero
engineering overhead. The paper also explores some techniques like caching to reduce the execution overload
of runtime dispatching.

Third, by the "stringly-typed" abstraction of operators, the proposed notation enforces more explicit
programming of the semantics of tensor axes, as well as more runtime checking to catch correctness issues.
This is particularly helpful among deep learning practitioners who are already exposed to a set of conventions
(e.g. "NCHW") to write the right code. The typed operators, instead of typed tensors (e.g. NamedTensor),
provides the guarantee that users only need to type important part of their code as a drop-in replacement of the existing
code fragments without having to worry about refactoring the entire codebase. This is again particularly helpful
for deep learning practitioners

**Weakness.** With the proposed notation, compiler-based approaches, e.g. TVM, MLIR, haven't been explored to further boost performance,
although such opportunities are real and tangible. For example in TVM, the Tensor Expression (TE) can be considered as
a generalized form of EinOps, and it could bring extra performance gain to the users. Therefore, it would be desirable to integrate
with those compiler backends to assist with potential performance.

The normal definition of the EinOps notation, while demonstrated in many examples, are not carried out formally.
For example, brackets `()` in the notation may serve different purposes under `rearange`, `reduce` and `repeat`,
on the left- and right-hand side of the notation. It may refer to tiling one dimension, or composition of new dimensions,
or removing a dimension, etc. Another example that needs more clarity is stack/concatenate where there are multiple
tensors involved, and might be desirable to state formally on the constraints implied in this particular case.

Last, while the notation is very flexible in expressing layout-related operations, it is yet under-explored on
expressing the common operators that deep learning researchers may care about, namely, many variants of convolution.

**Correctness.** The reviewer is not aware of any correctness issue in all demos and figures in this paper.

**Clarity.** The paper is overall rich in detailed and insightful explanation. There is some minor clarity issues as pointed out in the weakness section.



**Summary Of The Paper:**

Operators are key elements in deep learning (TensorFlow, PyTorch, etc) or scientific computing (NumPy) frameworks.
This paper introduces a generic, convenient and elegant way of massively describing large portion of frequently used operators
under a unified framework, EinOps, inspired by einsum, or the Einstein summation convention.
Under this convention, key components like multi-head attention in modern neural networks can be expressed
with fewer lines of code. The major contribution of this work includes:
- Presents a set of novel and highly expressive notations to represent hundreds of operators
in modern deep learning frameworks;
- Develops a library that interprets the proposed notations and offloads them to existing implementations
in NumPy or TensorFlow, PyTorch;
- Demonstrates significantly improved readability and debuggability with the proposed notations


**Summary Of The Review:**

In summary, the paper demonstrates clearly an elegant and novel notation to represent a large portion of operators in deep learning workloads, and developed a meta-framework with solid engineering. The reviewer recommends to consider the acceptance of the paper as a novel good deep learning meta-framework.

---

> ### Author Response · Authors · 2021-11-15
> **Response to reviewer c6LK**
>
> Thank you for this neat review.
>
> Summary and strength are so well-written that it feels tempting to steal some phrases.
>
> > With the proposed notation, compiler-based approaches, e.g. TVM, MLIR, haven't been explored to further boost performance, although such opportunities are real and tangible. For example in TVM, the Tensor Expression (TE) can be considered as a generalized form of EinOps, and it could bring extra performance gain to the users. Therefore, it would be desirable to integrate with those compiler backends to assist with potential performance.
>
> Agree, we'll mention this in the discussion section.
> Thanks for a specific pointer to TE, we'll investigate this opportunity for speeding up further.
>
> > The normal definition of the EinOps notation, while demonstrated in many examples, are not carried out formally. For example, brackets () in the notation may serve different purposes under rearange, reduce and repeat, on the left- and right-hand side of the notation. It may refer to tiling one dimension, or composition of new dimensions, or removing a dimension, etc. Another example that needs more clarity is stack/concatenate where there are multiple tensors involved, and might be desirable to state formally on the constraints implied in this particular case.
>
> Thanks for the feedback. We will add formal syntax and constraints in the revised version.
>
> > Last, while the notation is very flexible in expressing layout-related operations, it is yet under-explored on expressing the common operators that deep learning researchers may care about, namely, many variants of convolution.
>
> We agree with this.
>
> Researchers tend to use hardware-optimized operations (mostly, CuDNN) as they provide a solid combination of flexibility and performance.
> Thus, one either wraps CuDNN (and yet-another-way to describe convolutions is interesting as a concept, but not that valuable), or introduces notation for out-of-the-box operations (as e.g. Halide/TensorComprehension). Due to the low ratio of value over long-term engineering costs, we are unlikely to explore that direction.
> Some other DL operations (not convolutions) are a subject of future research.

---

### Author Response · Authors · 2021-11-15
**Response to all reviewers**

### Einops and Einsum

As we mention in the paper, einops notation was largely inspired by np.einsum and we tried to align interfaces for simultaneous usage. Appendix A demonstrates how those should interact together (einops documentation contains multiple examples).

On pages 6-7 there is numpy/einops mapping. Only 2 operations of 16 (transpose and swap axes) can be implemented with einsum.

Einops deviates from einsum in the following ways:

- Arbitrary reductions. Einops supports max-, min-, sum-, mean-, logaddexp- reductions (as well as callables to support any other reduction). Einsum can be used to perform sum-reduction only.
- Axes composition/decomposition. This core change is discussed in the main text with multiple examples
Multi-character names. Digits and underscores are allowed, arbitrary capitalization is allowed), space is used as a delimiter for axes names. In einsum "ij k -> k ij" transposes 3-dim tensor, while this is an operation on 2-dim tensors for einops.
- Unitary axes (in any pattern user can use "1" to reflect axis of size one)
- Specification of axes size, verification of shapes and divisibility.
- Anonymous axes. Axes that are present only on lhs or rhs can be specified with just size in the pattern. Example: repeat(x, "h w -> h w c", c=3) can be repeat(x, "h w -> h w 3"). Unitary axes are not subcase of anonymous axes.
einops.repeat (introduction of new axes on rhs by repeating elements along the axis). We demonstrate that it allows einops to support repeats and tiles.
- Backend-independence of einops notation. Einsum implementations suffer from non-aligned support for spaces between names, and capitals.
- List inputs. Einops can accept a list of tensors with identical shapes, which are stacked. E.g. rearrange([r, g, b, alpha], 'channel h w -> h w channel'). This extends einops to support tensor stacking and concatenations.
- Layers. einops functions have layer counterparts, e.g. einops.layers.torch.Rearrange('b c h w -> b (c h w)') can be used inside Sequential as one of the modules. In examples from pytorch documentation (otherwise sequential) modules are implemented using custom "forward" to accomplish this transformation with x.view or x.flatten.

We do not implement following features of np.einsum:
- No axes repeats on lhs. Einsum allows writing e.g. 'ii->' to get a matrix trace. To prevent coding mistakes, we dropped this. No user feedback argued for bringing it back.
- Multi-tensor operations. Einops focuses on single-tensor transformations. While there is an einops layer that operates on two tensors, the core functionality used by most users is one-tensor transforms.
Einops novelties (compositions, new axes, anonymous axes, unitary axes, specification of axes sizes) can readily be applied to extend e.g. einsum.
- Implicit indexing. It relies on letter sorting, does not encourage descriptive axes names as those define output axes order, and makes output implicit, not explicit; it also conflicts with einops.reduce. Github search shows negligible usage (<1%) of list-inputs syntax, and it was not added to einops.

### Venue placement

Formally, einops satisfies two topics mentioned in call for papers: implementation issues, software platforms.
Einops was submitted specifically for this track.

ICLR community is the right auditory for einops as these are (mostly) researchers heavily operating with tensors, and implementing novel models and algorithms.
Our solution would be most beneficial to them and can help in research on a daily basis.

### Validation of notation and implementation

Reviewers p5Km and gRMH push on statistical validation (user-study), which has extremely limited scope and value: neither it can guide long-term design of a set of dependent features, nor it helps to verify ability to smoothly integrate into existing codebase, nor it shows maintainability of resulting solutions, nor even proves ultimate usefulness.

Pytorch claims to be user-focused. Recent keynote talk by Pytorch lead (https://soumith.ch/posts/2021/02/growing-opensource/) confirms our position about poor relevance of artificial tests and ultimate validation by tracking usage dynamics, collecting and analyzing feedback.
We believe a couple of user studies will add little to no value to the reader, and can't be considered any stronger argument than adoption.

Einops similarly has taken a "test by time" approach. Open-sourced more than 3 years ago, it got adoption in public projects of most major AI labs, which is a strong indicator of novelty, usefulness and reliability.
While einops used studies on small user groups to validate its design, there is little value that can be added to adoption evidence at this stage.
We want to highlight the paper's focus: analyze previous ideas in detail, describe our approach, technical challenges and solutions, and explain which problems a researcher can expect to be resolved. The latter is supported by examples and implementations for real-world cases.

---

### Author Response · Authors · 2021-11-23
**Rebuttal revision**

Dear Reviewers and Area Chairs,

Thanks again for your constructive and specific feedback which helps us to improve clarity and presentation of the paper.

We have incorporated fruitful suggestions on improving the paper's text and included additional information requested by reviewers. The main changes made in the text are:
- The formal grammar of proposed notation is given in Section 4 along with its detailed description and features
- As the issue on relation between einops and einsum was raised by several reviewers, we add it in Sections 2 and 4 with detailed comparison in Appendix B
- Due to space constraints we included a user study for the prototype stage of einops in Appendix C, which demonstrates intuitiveness and readability of the main design choice for the notation
- Einops overhead on top of the DL frameworks is provided in Appendix F for 3 use cases and different input sizes
- Details on caching mechanism and why it is important is given in Appendix G
- We moved discussion of related work on labeled tensors approach and convolutions into Appendix A
- We describe initial confirmation of einops applicability to a wide range of DL applications and architectures in Appendix H
- Finally, validation of einops suitability by adoption is covered in Appendix I
- We also revisited Discussion section
- Multiple minor changes addressing reviewers concerns

Let us know if you have any other questions. Thank you!

---

### Decision · Program_Chairs · 2022-01-20

**Decision:**

Accept (Oral)

**Comment:**

All reviewers agree that this paper is a useful and valuable contributions to ML engineering.
 - insightful analysis .. highly user friendly operator design
 - "useful and I can see it having large adoption in the community of scientific computing" ... "
 - "Personally I tend to buy these advantages of einops" ... "However, there is a lack of solid empirical study to validate the effectiveness and efficiency of the design"
 - "a useful and appealing new coding tool."

The negative reviewers appear fixated on the (true) observation that the paper does not look like a conventional ICLR paper, thati it "reads like a technical blog", and "lacks rigour".
I belive it is fair and measured to state that these reviews may be considered to exhibit aspects of gatekeeping: requiring more "mathiness" that does not help the paper, or more "rigour" through user studies that are in fact less valuable than the reviewers' own observations "I could see myself...", "I tend to buy...".

This is a paper about design, not about models or algorithms (although the algorithmic work is good).  It is about the design of tools that we all use, and about the decisions and thought processes that led to that design.  A reviewer decries "many non-rigorous claims".  These are claims about the usability of existing systems, and mostly appear in the discussion and footnotes, as the authors note in rebuttal.  Of course, one could have run user studies to back up each claim, but I am just as convinced by the examples shown in the paper.  It matters not to me what some users corralled into a user study thought.  It matters what I and my colleagues will think, and I am now sure to recommend einops to colleagues.  I would not have met it had the paper not been submitted to ICLR, and hence I am certain it should be accepted, so more can see that we care not just about mathiness, but actually enabling progress in our field.

The job of a conference like ICLR is to expose researchers and practitioners in machine learning to ideas and techniques that may advance their research and practice.  Programming, and the translation of mathematical ideas to efficient computer code, are fundamental to all of machine learning, and hence programming models are very much suitable for presentation to an ICLR audience.  I am certain that this paper, and the technology it describes, are more important to ICLR readers than knowing that if module A is co-trained with module B, then combined with compression C, the SOTA on some arbitrary benchmark is increased by 0.31 +/- 1.04.

Reviewer gRMH says "there is no code", but the code has been in the open for three years; it is an accident of our misapplication of the principles of blind review that the reviewer felt they could not search for the code, and that the authors felt they could not bring to bear the evidence that three years of real-world usage have brought.

Reviewers say the work is just an extension of einsum, while noting that the extension is useful and nontrivial.  Yes, it is an extension, and the paper's examples show how it yields more compact code that is also more readable and maintainable.

I could add more examples, but in short, I tend to side with the authors' response at almost every point.  At the same time, the final version of the paper has been strengthened by this dialectic, and I expect further strenghtening through exposure to the ICLR community.

To the authors: Listing 1 is useful, but should be in an appendix.  Instead, add examples of ellipses on P5, and show more inline examples in general.  The paper would be strengthened by another pass over the English -- after the decision is made I would be happy to volunteer to help.